# Green synthesis and characterization of zinc chitosan nanoparticles with their anti-bacterial study against rice pathogen *Xanthomonas oryzae* pv. *oryzae*

Lutfur Rahman[1], Rumana Akter[1], Md. Omar Kayess[1], Rakibul Hasan[1],
A. K. M. Sahfiqul Islam[1], Sheikh Arafat Islam Nihad[1], A. K. M. Mohiuddin[2], A. L. Nayeem[1],
Mohammad Abdul Latif [1]*

1 Plant Pathology Division, Bangladesh Rice Research Institute (BRRI), Gazipur, Bangladesh,
2 Department of Biotechnology and Genetic Engineering, Mawlana Bhashani Science and Technology University (MBSTU), Santosh, Tangail, Bangladesh

☯ These authors contributed equally to this work.
* alatif1965@yahoo.com

## Abstract

Bacterial leaf blight (BLB) of rice is caused by the bacterium *Xanthomonas oryzae* pv. *oryzae* (*Xo*o) and leads to significant yield loss. The overuse of antibacterial chemicals casue environmental toxicity, high cost, and low biosafety generating interest in eco-friendly alternatives.This study evaluated the antibacterial efficacy of biosynthesized zinc–chitosan nanoparticles (ZnChNPs) against *Xanthomonas oryzae* pv. *oryzae* (*Xoo*). Successful synthesis was confirmed by UV–visible spectroscopy with a characteristic absorption peak at 356 nm. Comprehensive physicochemical characterization using scanning electron microscopy (SEM), transmission electron microscopy (TEM), and field emission scanning electron microscopy (FESEM) revealed ZnChNPs with diverse polymorphic morphologies, while energy dispersive spectroscopy (EDS) confirmed zinc, oxygen, and carbon as the major elemental components. Fourier transform infrared spectroscopy (FTIR) analysis identified key functional groups associated with nanoparticle formation and stabilization, and X-ray diffraction (XRD) patterns verified a crystalline hexagonal wurtzite structure. Dynamic light scattering (DLS) measurements showed a mean hydrodynamic diameter of 256.2 nm, and a zeta potential of −25.1 mV indicated good colloidal stability of the formulation. *In vitro*, ZnChNPs:dH$_2$O at 8:2 and 10:0 ratios showed inhibition zones of 22.75 mm and 24.84 mm, respectively. In net house conditions, treatments reduced BLB lesion lengths by 36%–73.54% at 14 days and 34.98%–63.66% at 21 days. The Bismerthiazol (0.15%) + ZnChNP:dH$_2$O (8:2) treatment reduced lesions by 68.28% and 55.48% at 14 and 21 days, respectively. Field trial results showed 66.14% and 61.27% reductions at 14 and 21 days. These findings suggest ZnChNPs can be used as a sustainable strategy for controlling BLB in rice in an effective manner.

**Data availability statement:** All data are in the manuscript and/or Supporting information files.

**Funding:** Sustainable management of blast, sheath blight and bacterial blight diseases of rice through Nanoparticles (Project ID:TF 71-C/20) funded by Krishi Gobeshona Foundation (KGF), Ministry of Agriculture, Bangladesh.

**Competing interests:** The authors have declared that no competing interests exist.

## Introduction

About half of the world's population derives its daily energy needs from rice, which is the most widely consumed grain in the Poaceae family [1–3]. Bangladesh is the 3rd largest rice producer in the world, where the security of rice is a prerequisite for food security [4,5]. According to projections, 58–567 million tonnes (Mt) of rice will need to be produced in 2030 to fulfil the global requirement. Till now 32 rice diseases have been documented in Bangladesh. According to disease severity and incidence, bacterial leaf blight, blast, sheath blight, tungro, and false smut are considered major diseases throughout the country [6–9]. Among them, BLB is currently a major threat to both inbred and hybrid rice farming worldwide, particularly in Southeast Asia, especially Bangladesh [10–14]. *Xoo* causes BLB, resulting in yield losses ranging from 10–50% in the southeast and southern Asian countries, with potential losses of up to 80%. The extent of crop loss is influenced by different factors such as crop varieties, growth stage, geographical location, and environmental conditions [15–17]. As a result of its extensive impact on rice production and its complex pathogenic nature, *Xoo* is consistently recognized as one of the ten most economically significant plant pathogenic bacteria worldwide [18]. In field conditions of Bangladesh, a yield loss of around 5.8–30.4% was due to BLB disease [15]. The severity of damage from BLB disease varies based on factors such as location, virulence of the race, crop growth stage, weather conditions, and cultivar types [19,20]. The meteorological conditions, including temperature, humidity, flooding, rainfall, and stormy weather, can potentially affect the disease prevalence [5,21,22]. To fight this disease, chemical control techniques is not being suggested [23], due to their environmental toxicity. Plant bacterial infections cause significant global losses annually, and while most countries have banned agricultural antibiotics like streptomycin and oxytetracycline to curb antibiotic resistance, others have escalated their use.

The management of BLB has become increasingly complex due to the rapid evolution and genetic diversification of *Xoo* populations, a process largely driven by host–pathogen co-evolution and intensified agricultural practices [24,25]. Genomic and pathotypic studies have identified multiple highly virulent lineages, including strains capable of circumventing resistance genes that are widely used in commercial rice varieties [25,26]. Although the development of resistant cultivars remains a cornerstone of BLB control strategies, the effectiveness of this approach is frequently compromised by the continual emergence of novel *Xoo* pathotypes and the organism's high genomic plasticity [24,27,28]. Additionally, the introgression of resistance genes into elite cultivars often results in adverse effects on key agronomic traits, creating a trade-off between disease resistance and yield or grain quality [29]. Furthermore, the strong selective pressures imposed by resistance breeding programs may inadvertently accelerate the evolution and spread of virulent *Xoo* strains [24].

In rice disease management, ZnONPs have shown promising antimicrobial activity, initially against fungal pathogens and more recently against *Xoo*, the causal agent of bacterial blight [30–34]. Emerging evidence suggests that ZnONPs exert their antibacterial effects on *Xoo* primarily through mechanisms involving oxidative stress induction and disruption of bacterial membrane integrity, positioning them as potential

next-generation bactericides for BLB control [31–34]. While intracellular reactive oxygen species (ROS) production has been observed in *Xoo* cells following nanoparticle exposure, existing studies have predominantly focused on laboratory strains, with limited attention to highly virulent, field-derived isolates [34,35]. Consequently, critical knowledge gaps remain regarding the nanoparticle-pathogen interactions under realistic pathogenic conditions, underscoring the need for comprehensive mechanistic studies involving aggressive *Xoo* populations [35].

Chitosan is one of the most plentiful materials with various uses in food packaging, drinks, and in agriculture [36]. Due to its minimal toxicity, excellent biocompatibility, and ability to biodegrade naturally, chitosan has gained significant attention as a fundamental structural element in the development of nanomaterials. For instance, a chitosan-TiO2 nanocomposite with antibacterial activity against *Xoo* was produced and described by [37]. However, chitosan polymers and nanoparticles possess antimicrobial activity due to their electric charge, high adsorption, and chemical reactions that allow them to interact effectively with the bacterial cell membrane [38]. Their ability to inhibit bacterial growth is influenced by their size, shape, biological characteristics, and structural features.

ZnO NPs can be synthesized through physical, chemical, or biological approaches [39]. Although physical and chemical methods typically produce nanoparticles with uniform size and high purity, they frequently require toxic chemicals, substantial energy consumption, and present environmental risks [39,40]. Conversely, green synthesis employing plant extracts has emerged as a sustainable, cost-effective, and environmentally benign alternative [40,41]. The bioactive constituents of plant extracts—including flavonoids, terpenoids, and saponins—function as natural reducing and stabilizing agents during the nanoparticle synthesis process [40–43].

Zinc significantly contributes to the optimal development of plants by engaging in numerous enzymatic bioreactions and various metabolic processes. Besides, numerous enzymes utilize this micronutrient as a regulatory cofactor, such as oxidoreductases, isomerases, hydrolases, transferases, and ligases [44]. In addition, tomatoes possess a substantial quantity of ascorbic acid, polyphenols, particularly flavonoids [45], which render them a promising material for the synthesis of nanoparticles.

In recent times, an increasing number of researchers have provided evidence that metal nanoparticles, including Ag, ZnO, $TiO_2$, MgO, and CuO, in addition to the natural biopolymer chitosan and its derivative, exhibit potent antibacterial properties against a wide range of bacterial pathogens [46]. In comparison, earlier studies demonstrated that biosynthesized magnesium oxide (MgO) nanoparticles also possess notable antimicrobial properties, highlighting their potential as alternative agents for controlling plant pathogenic bacteria [47] Similar findings were also obseved for Ag nanoparticles [48] and ZnO nanoparticles [47] that inhibit the growth of the *Xoo* pathogen [30,31]. Despite extensive studies on zinc-based nanoparticles and chitosan-derived materials as individual antimicrobial agents, there is a lack of research on green-synthesized zinc–chitosan nanocomposites and their targeted antibacterial activity against *Xanthomonas oryzae* pv. *oryzae*. In particular, the relationship between eco-friendly synthesis, physicochemical characteristics, and antibacterial performance of such nanocomposites remains insufficiently explored.

This research aimed to develop a simple, cost-effective, and ecofriendly method for the biosynthesis zinc chitosan nanoparticles with effective antibacterial properties against *Xoo* under both *in vitro* and *in vivo* conditions.

## Materials and Methods

### Experimental materials

Fresh red tomatoes were obtained from the local market near the Bangladesh Rice Research Institute, Gazipur, Bangladesh. The chemicals used in this study, like zinc sulfate heptahydrate ($ZnSO_4 \cdot 7H_2O$), low molecular weight chitosan powder, sodium tripolyphosphate (STPP), and high-purity sodium hydroxide, were sourced from Sigma-Aldrich (St. Louis, MO, USA).

**Formulation of aqueous extract of tomato.** Tomato extract was utilized as a natural reducing agent for the green synthesis of NPs, primarily due to its cost-effectiveness, extensive availability, and intrinsic bioactive compounds. To

prepare the extract, fresh tomatoes underwent thorough washing under running tap water to eliminate surface impurities and organic debris, followed by several rinses with distilled water. Afterward, the cleaned tomatoes were oven-dried at 35°C. Approximately 10 grams of the dried tomato material were then combined with 200 mL of distilled water in a clean beaker. This mixture was then heated at 80°C for 4 hours while being continuously stirred on a hot plate with magnetic stirring. The resulting solution was double-filtered using the Whatman No. 1 filter paper to ensure optimal clarity. The freshly prepared aqueous extract was immediately utilized for the synthesis of ZnChNPs, while the residual portion was stored at 4°C for future use.

## Eco-friendly synthesis of zinc–chitosan nanoparticles

According to the combined methods of [49] and [31] ZnChNPs were synthesized with slight changes (Fig 1). For the green synthesis of ZnChNPs, 50 mL of 0.25 M zinc sulfate heptahydrate ($ZnSO_4 \cdot 7H_2O$) aqueous solution was gradually added to 100 mL of tomato extract. The reaction mixture was maintained under constant magnetic stirring at 300 rpm and 60°C for 1 hour. The pH of the resulting suspension was adjusted to 12 using 4 M sodium hydroxide (NaOH) to facilitate nanoparticle formation. Separately, a 0.75% (w/v) chitosan solution was made by dissolving 1.5 g of low molecular weight chitosan in 200 mL of 2% acetic acid, followed by continuous magnetic stirring at 60°C for 2 hours to confirm complete dissolution and activation of chitosan as a stabilizing agent. To synthesize ZnChNPs, the prepared chitosan solution was introduced into the zinc-tomato reaction mixture, followed by the addition of sodium tripolyphosphate (TPP) as a crosslinking agent following a modified method of [31,49]. Subsequently, 4 M sodium hydroxide (NaOH) was gradually added dropwise to the reaction mixture to adjust the pH to 12, while maintaining continuous stirring at 80°C for an additional 2 hours. This step was essential to promote effective crosslinking and facilitate the successful formation of nanoparticles as shown in Fig 1. The final nanoparticle suspension was subjected to centrifugation at 10,000 g for 20 minutes to remove the supernatant. The resulting pellet was thoroughly washed using distilled water and ethanol to eliminate residual impurities and unreacted materials. The purified ZnChNPs were then oven-dried at 60°C to yield the final nanoparticle powder as described by [31].

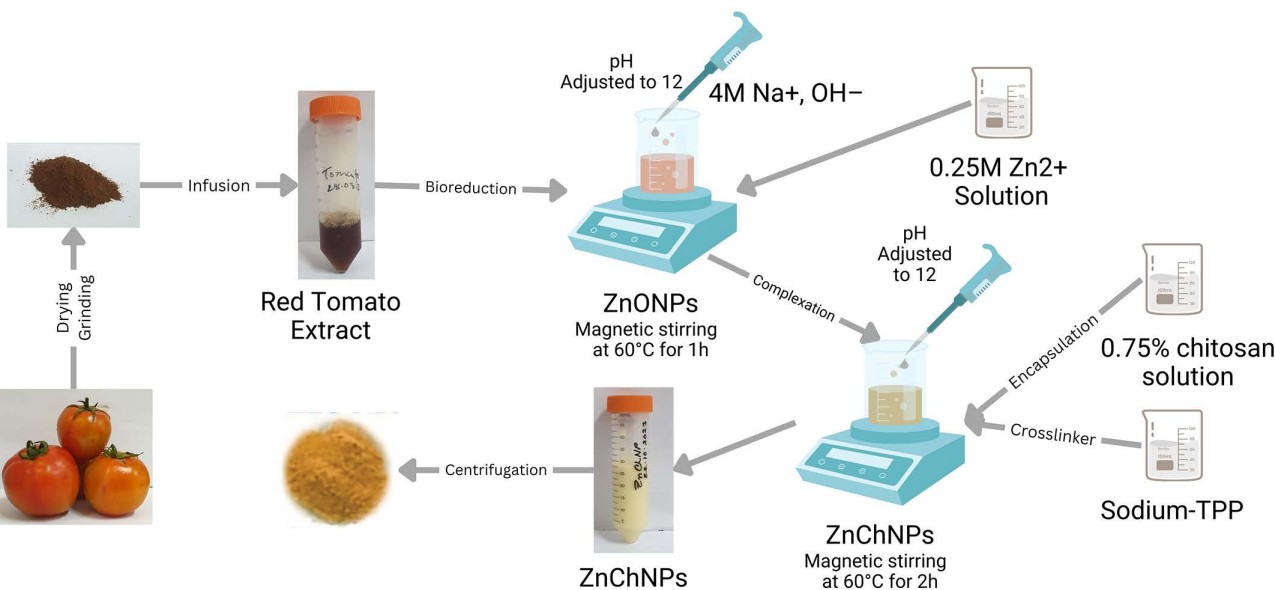

**Fig 1. Biosynthesis and formation of ZnChNPs from tomato extract.** Illustration showing the transformation of red tomato extract into ZnChNPs. The schematic highlights the key stages, including nanoparticle nucleation, chitosan encapsulation, and crosslinking, resulting in the formation of stable ZnChNPs.

## Characterization of ZnChNPs

**UV-Vis spectroscopy based assessment.** The peak absorption levels of ZnChNPs were periodically recorded through UV–vis spectroscopic analysis using a double-beam UV–vis spectrophotometer (Shimadzu UV-2600i, Japan) within the 200–800 nm range to evaluate the optical properties of the biosynthesized nanoparticles..

**SEM and FESEM based observation.** The surface morphology of the synthesized nanoparticles was meticulously examined using SEM. This analysis was performed with an accelerating voltage of 15 kV. To enhance electrical conductivity and optimize imaging, the sample was first coated with a thin layer of gold. The SEM analysis itself was conducted using a JCM-7000 model located at the Plant Pathology Division of the BRRI. Furthermore, high-resolution surface imaging was carried out employing FESEM. This advanced analysis utilized a JEOL-7600F instrument, situated at the Department of Nanomaterials and Ceramic Engineering, Bangladesh University of Engineering and Technology (BUET), Dhaka-1000.

**Analysis of FTIR.** In this study, Fourier Transform Infrared (FTIR) spectroscopy was used to determine the functional groups present in ZnChNPs. The analysis was conducted at the Fibre & Polymer Research Division, BCSIR Laboratories, Dhaka, using a Perkin Elmer Spectrum II FTIR spectrometer. The dried, biosynthesized ZnChNPs were scanned from 500 to 4000 $cm^{-1}$ with a resolution of 4 $cm^{-1}$, following the procedure outlined by Abdallah et al. [31].

**Analysis of XRD.** The crystallographic characteristics of the synthesized ZnChNPs were investigated using XRD analysis as descrbed by Abdallah et al. [31]. This was performed with a Smart Lab X-ray diffractometer (Rigaku, Japan) at the Institute of Glass and Ceramic Research & Testing under the Bangladesh Council of Scientific and Industrial Research (BCSIR), Dhaka, Bangladesh. The X-ray source operated at 40 kV and 50 mA, utilizing Cu Kβ radiation with a wavelength of 0.154 nm. Diffraction patterns were collected within an angle of 2θ and range of 5° to 90°, at a scanning speed of 30.0°/min, to determine the crystalline structure and phase composition of the nanoparticles. The crystalline size was determined from the XRD data using the Scherrer equation [50].

**Particle size and zeta potentials distribution.** The particle size distribution and zeta potential of the synthesized ZnChNPs were measured using a Litesizer 500 (Anton Paar, United Kingdom). Before analysis, an aliquot of the nanoparticle suspension was diluted with ultra-purified water to ensure optimal dispersion. The diluted sample was then subjected to sonication for 15 minutes to minimize agglomeration and achieve uniform particle dispersion. These measurements provided critical insights into the hydrodynamic diameter and colloidal stability of the ZnChNPs.

**Observations based on TEM-EDS.** The morphology of the synthesized ZnChNPs was examined using TEM with a Talos F200X G2 instrument (Thermo Fisher, Czech Republic). The analysis was done at the Materials Science Division of Bangladesh Atomic Energy Commission, Dhaka, Bangladesh. For sample preparation, a thin film was prepared by dropwise application of a small amount of nanoparticle suspension onto a carbon-coated copper grid. The grid was then dried under a mercury lamp for 5 minutes to ensure proper adhesion and solvent evaporation. Then TEM imaging allowed for high-resolution visualization of the nanoparticle shape and distribution.. Then, TEM imaging allowed high-resolution visualization of nanoparticle shape and distribution. Additionally, the instrument was equipped with an Energy Dispersive EDS system, which was used to confirm the elemental composition and verify the presence of the nanoparticles.

## Antibacterial Activity

***In vitro* antibacterial effects of ZnChNPs and bacterial growth.** This *in vitro* study investigated the efficacy of ZnChNPs at various ratios against microbial growth compared to control treatments. To evaluate *the in vitro* antibacterial effects of ZnChNPs against *Xoo,* five different concentration ratios (10:0, 8:2, 6:4, 4:6, 2:8) of ZnChNPs prepared from tomato extract were tested using a well diffusion method which is described by Perez et al. [51]. In this experiment, tomato extract, chitosan (0.75%), 0.25M zinc sulfate salt ($ZnSO_4.7H_2O$), and sterilized water were used as controls. For all cases, 3 replications were maintained. A total of 200 μL of overnight bacterial culture ($1 \times 10^8$ CFU $mL^{-1}$) was mixed with 5 mL of

 

PSA medium. Then, the bacterial culture was streaked on PSA (Peptose Sucrose Agar) media. The different treatment ratios and controls were applied as 10 μL aliquots into wells on the inoculated plates using a pipette. The inoculated plates were incubated at 28°C to 30°C for 21 days.. After full growth in the plates, the radial growth of each culture was measured at **24 hours and 48 hours** to assess the antibacterial efficacy of ZnChNPs. **Measurements at 7 and 21 days were excluded, as *Xoo* growth becomes static beyond 72 hours in agar-based assays and does not reflect active antibacterial response.**. The zone of inhibition was measured using a physical ruler, such as a meter scale. The ruler was placed directly above the Petri dish, and the diameter of the inhibition zone was determined through visual observation. Using the equation *diameter* = $\sqrt{(\pi \times Area)/4}$, the diameter of the inhibition zone, was calculated, and the well diameter was subtracted to obtain the exact bacterial growth inhibition zone.

***In vivo* examination of ZnChNPs on bacterial leaf blight disease at net house conditions.** For the pot experiment, 40 pots were prepared and filled with air-dried soil. The rice variety BRRI dhan48 was selected for this study. Seeds were initially sown in the seedbed of the Plant Pathology Division at BRRI. Following a 30-day period, healthy seedlings were transplanted into prepared pots, accommodating three seedlings. To ensure the reproducibility and statistical robustness of the findings, all experimental treatments were conducted in three replications. For inoculation, the leaf tips of the seedlings were trimmed at a slight diagonal using sterilized scissors dipped in the bacterial suspension. This procedure was carried out at the maximum tillering stage. The seedlings were then observed for the onset of BLB symptoms using freshly prepared BLB isolates at $1 \times 10^8$ CFU mL$^{-1}$ concentration. Each treatment received two foliar sprays (prepared ZnChNPs) one on the second day and another on the seventh day following inoculation. Lesion length was measured after 14 and 21 days to assess treatment efficacy, and data on the BLB disease's development were appropriately documented along with the dates of the disease's onset and incidence. To investigate the efficacy of ZnChNPs alone as well as formulations combined with Bismerthiazol (Bis) (3 dosages; 0.15%, 0.125%,.0.1%) at varying ratios (8:2 and 6:4) for reducing lesion length in BLB disease under net house conditions. The effectiveness was compared to Bismerthizol(0.2%) and ZnChNPs(10:0) treatment, a healthy control, and a disease control. A total of ten treatments were evaluated for their efficacy against bacterial leaf blight (BLB) under net house conditions, as follows: T1: healthy control; T2: disease control; T3: (ZnChNPs; T4: bismerthiazol at 0.2%; T5: bismerthiazol (0.15%) combined with ZnChNPs and distilled water at an 8:2 ratio; T6: bismerthiazol (0.125%) with ZnChNPs and distilled water at 8:2; T7:, bismerthiazol (0.1%) with ZnChNPs and distilled water at 8:2; T8: bismerthiazol (0.15%) with ZnChNPs and distilled water at 6:4; T9: bismerthiazol (0.125%) with ZnChNPs and distilled water at 6:4; and T10: Bismerthiazol (0.1%) with ZnChNPs and distilled water at 6:4. Treatments efficacy was evaluated by calculating the percentage reduction in lesion length relative to the disease control.

***In vivo* effect of ZnChNPs on bacterial leaf blight under field conditions.** The experiment was performed in Gazipur at the BRRI's Plant Pathology Division (23°59′20″ N, 90°24′28″ E) during the T. Aman session, 2023. The rice variety BRRI dhan51 was employed as the test material in this experiment. The experimental plots measured 3 meters by 1 meter. A randomized complete block design (RCBD) with three replications was employed to ensure the statistical reliability of the experiment. Subsequently, 30-day-old seedlings were transplanted with a spacing of 20 cm by 20 cm, in accordance with standard cultural practices. Inoculation was performed at the maximum tillering stage by trimming the leaves at a slight curved angle using scissors dipped in a virulent bacterial isolate. The inoculated plants were then monitored for the progression of BLB disease.The same treatments of the net house experiment were sprayed 3 days before inoculation for preventive measurement, and the other two sprays were applied after 3 days and 7 days of inoculation for curative measurement. Lesion length was measured after 14 and 21 days, and data were collected and appropriately documented along with the dates of the disease's onset and incidence at 14 days and 21 days after inoculation.. To prevent unintended dispersal of treatments to non-target plants, nanoparticle applications were carried out using controlled, low-pressure spraying under calm weather conditions. The effectiveness of the treatments was determined by calculating the percentage reduction in lesion length relative to the untreated disease control. Furthermore,

 

monthly environmental data, including temperature, relative humidity, and rainfall, were documented during the field trials to characterize the prevailing climatic conditions at the experimental site (S1 Table).

## Statistical analysis

Data from different *in vitro* and *in vivo* time periods were analyzed using one-way analysis of variance (ANOVA) with the Statistical Tool for Agricultural Research (STAR, IRRI). Mean comparisons were performed using the least significant difference (LSD) post hoc test ($p = 0.05$). Additionally, standard deviation and standard error values were calculated utilizing Microsoft Excel.

## Results

### UV-vis confirmation of biosynthesized ZnChNPs

The biosynthesis of nanoparticles using tomato extract is outlined in Fig 1. The Fig 2 presents the UV-visible absorbance spectra of tomato extract, synthesized ZnONPs, and ZnChNPs. The ZnONPs exhibited a characteristic absorption peak at 306 nm, while the tomato extract showed peaks at 212 nm and 287 nm. Following the incorporation of ZnONPs into the chitosan-TPP mixture, synthesis was confirmed by UV–visible spectroscopy at 356 nm.

### ZnChNPs characterization

The ZnChNPs were characterized using SEM, FESEM, and TEM-EDS (Transmission Electron Microscopy coupled with Energy Dispersive X-ray Spectroscopy), alongside FTIR, XRD, DLS, and zeta potential analyses to ensure their effective application in the experiment. SEM and FESEM analyses revealed that the synthesized ZnChNPs exhibited diverse polymorphic morphologies, including spherical and hexagonal shapes, with particle sizes ranging from 81.3 to 256.9 nm (Fig 3). TEM analysis further confirmed the presence of uneven and polymorphic nano-aggregate structures, suggesting regions of enhanced electrostatic attraction within the nanoparticles, with sizes broadly distributed between 100 and 300 nm, while EDS analysis verified the elemental composition of the ZnChNPs, showing zinc (29.46%), oxygen (41.41%), and carbon (6.48%) as the major constituents (Fig 4). FTIR spectral analysis displayed characteristic absorption bands at 3394, 2922, 1626, 1486, 1384, 1261, 1107, 1065, 1007, 875, and 467 cm$^{-1}$, indicating the presence of functional groups likely involved in nanoparticle formation and stabilization (Fig 5). XRD analysis confirmed the crystalline nature of the ZnChNPs, with distinct diffraction peaks at 2θ values of 31.86°, 34.55°, 36.39°, 47.67°, 56.77°, 62.89°, 66.62°, 68.14°, and 81.42°, corresponding to the (100), (002), (101), (102), (110), (103), (112), (220), and (311) crystallographic planes, consistent with an uneven polygonal crystalline structure (Fig 6). DLS measurements showed particle diameters ranging from

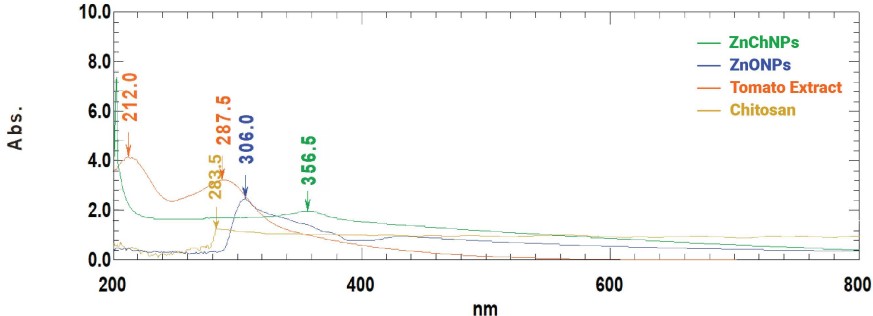

**Fig 2. UV-Vis absorption spectra of synthesized ZnChNPs and precursor materials.** The absorption profiles are displayed for ZnChNPs (green), ZnONPs (blue), Tomato Extract (orange), and Chitosan (yellow) over a wavelength range of 200 to 800 nm.

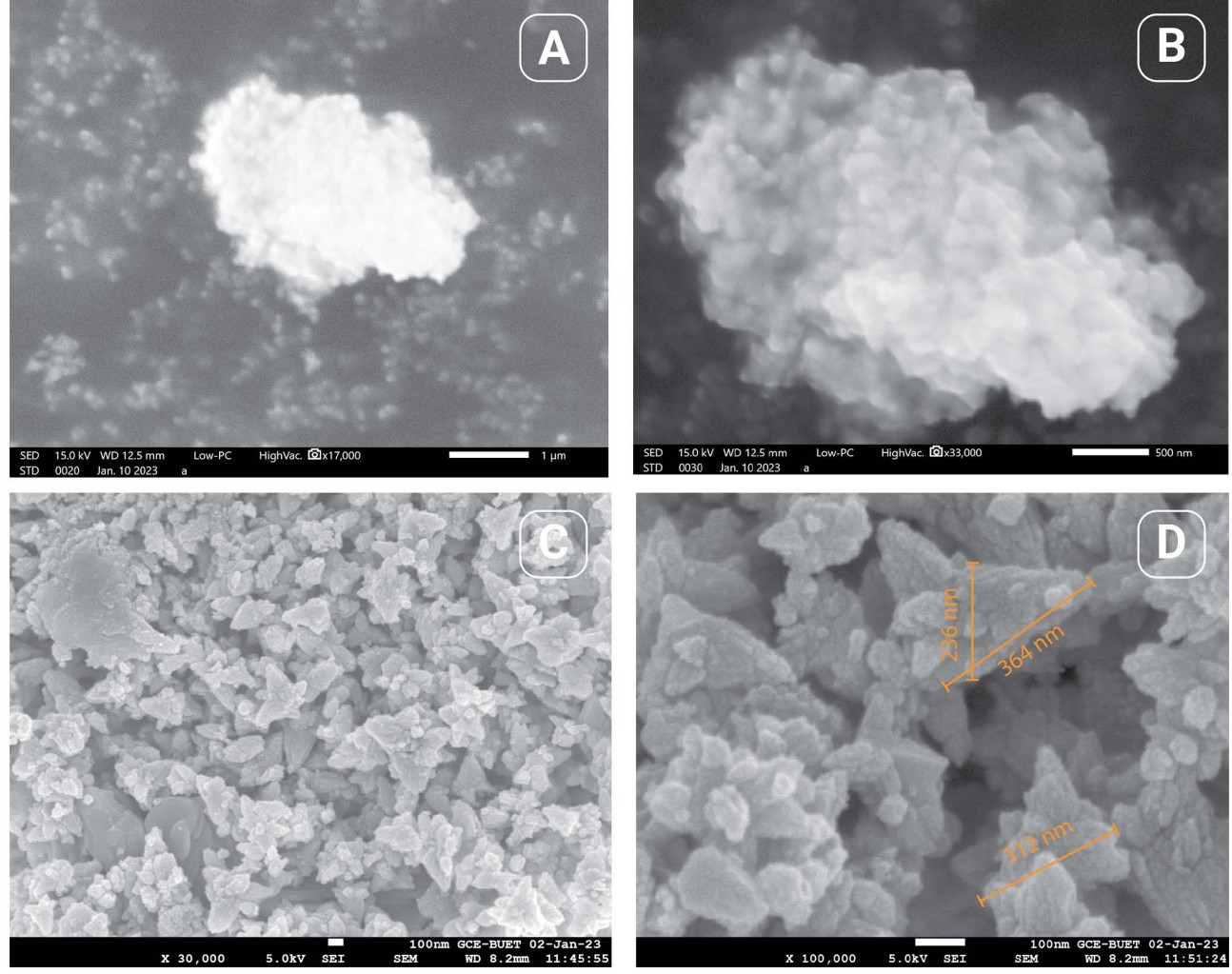

**Fig 3. Morphological characterization of bio-synthesized ZnChNPs. (A, B)** Field Emission Scanning Electron Microscopy (FESEM) images showing surface morphology and particle shape; **(C, D)** Scanning Electron Microscopy (SEM) images depicting overall particle size and aggregation patterns.

82.3 to 861.5 nm with an average size of 256.2 nm and a PDI of 0.25, indicating a moderately uniform size distribution, while zeta potential analysis revealed a value of −25.1 mV at pH 12.0, reflecting good colloidal stability of the ZnChNP suspension (Table 1) (Fig 7A, 7B).

### *In vitro* performance of ZnChNPs against bacterial leaf blight disease

All ZnChNPs treatment ratios exhibited significantly stronger inhibitory effects against microbial growth compared to the tomato extract and chitosan controls. Among these, the ZnChNPs:dH$_2$O 10:0 ratio demonstrated the largest zone of inhibition, reaching 24.84 mm at 48 hours after incubation. Fig 8 illustrates the comparative antimicrobial activity of various ZnChNPs:dH$_2$O ratios against microbial growth during the 24–48-hour evaluation period. The 8:2 ratio consistently inhibited microbial growth, producing zones of inhibition ranging from 22.37 mm to 22.75 mm within 48 hours. In contrast, the 6:4, 4:6, and 2:8 ratios exhibited progressively smaller inhibition zones, corresponding to the decreasing ZnChNPs

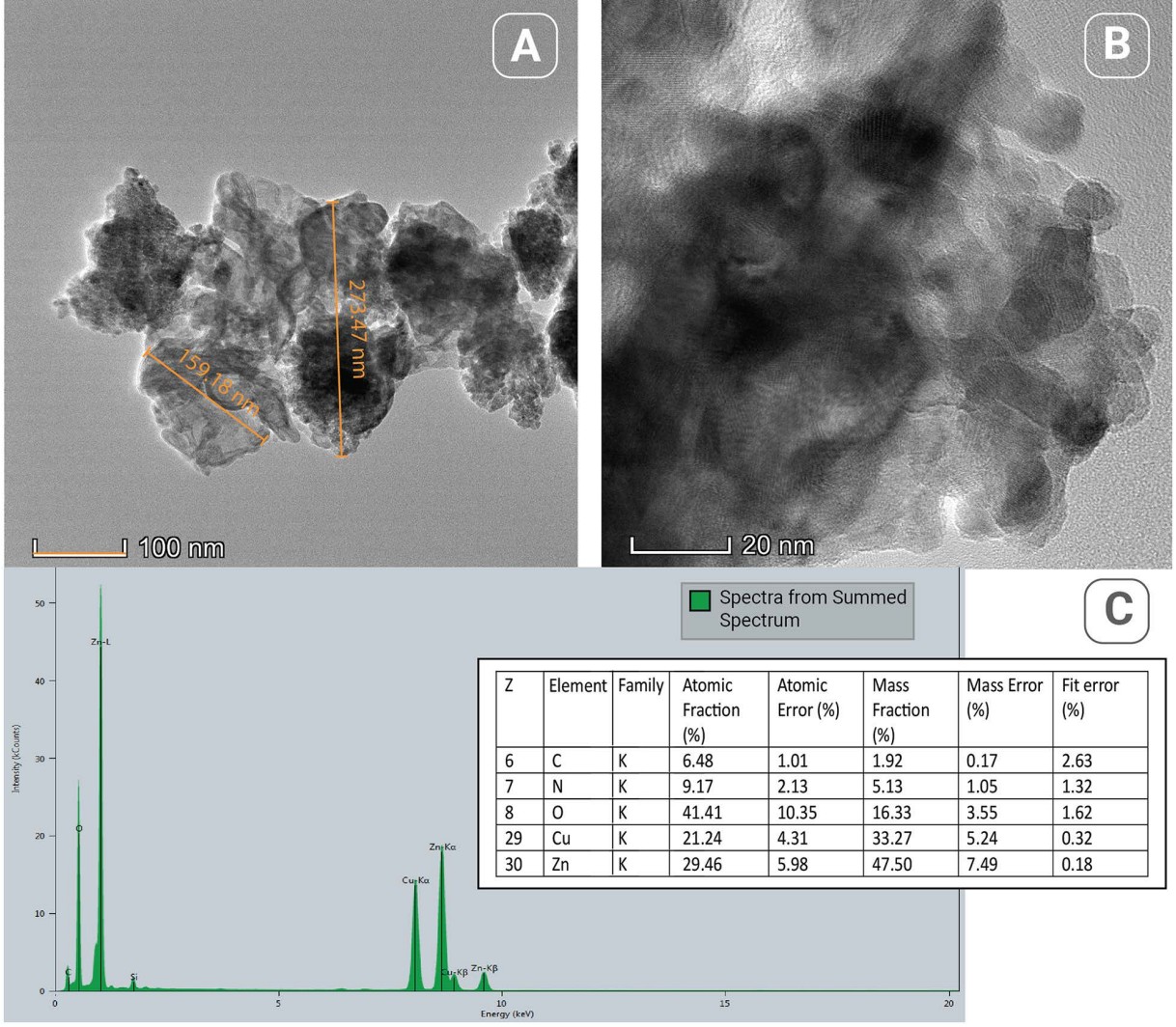

**Fig 4. Morphological and elemental characterization of bio-synthesized ZnChNPs. (A, B)** Transmission Electron Microscopy (TEM) images at different magnifications illustrating the morphology and size distribution of the nanoparticles. **(C)** Energy Dispersive X-ray Spectroscopy (EDS) spectrum and corresponding elemental data. The spectrum confirms the presence of Zinc (Zn) as the primary metallic constituent. The detection of Carbon (C), Nitrogen (N), and Oxygen (O) peaks suggests the presence of biological molecules likely proteins or phytochemicals. The accompanying table summarizes the atomic and mass fractions, confirming the chemical identity and purity of the sample.

concentration, with averages between 12.43 mm and 19.85 mm throughout the experiment. Although $ZnSO_4 \cdot 7H_2O$ (control) showed the strongest inhibition overall, with zones consistently between 31.83 mm and 33.18 mm at 24 and 48 hours, its high salt concentration renders it ineffective against bacterial leaf blight in rice. Tomato Extract exhibited the least inhibitory effect, with a zone of inhibition of 11.61 mm at 24 hours, slightly increasing to 12.95 mm by day 21. The 0.75% chitosan solution displayed a moderate inhibition zone, measuring 19.68 mm at 24 hours, which gradually decreased over time. As expected, the negative control (distilled water only) showed no inhibitory effect throughout the experiment. Measurements beyond 48 hours were not considered for analysis, as *Xoo* growth stabilized and no further significant change in inhibition zones was observed.

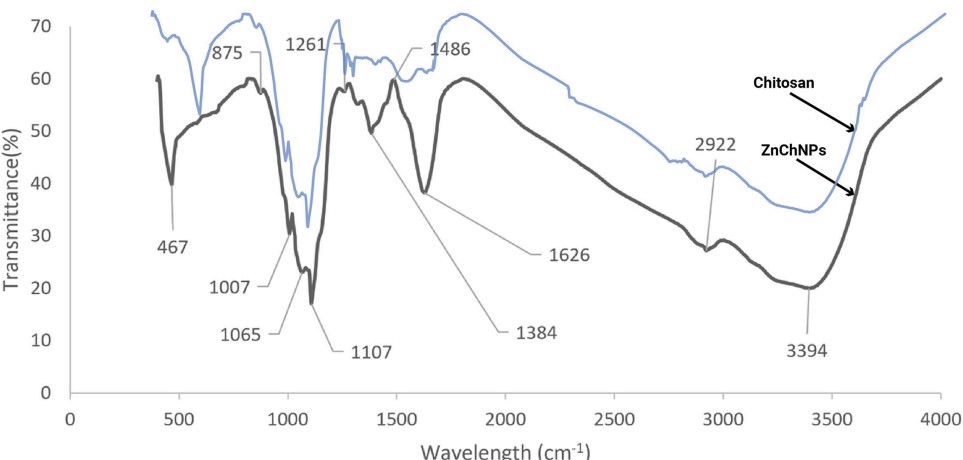

**Fig 5. FTIR spectral comparison of pure Chitosan and ZnChNPs.** Fourier-transform infrared (FTIR) spectra of pure chitosan (blue line) and ZnChNPs (black line). The comparison highlights change in characteristic vibrational bands and functional group interactions, confirming the formation of nanoparticles and the involvement of chitosan functional groups during zinc incorporation.

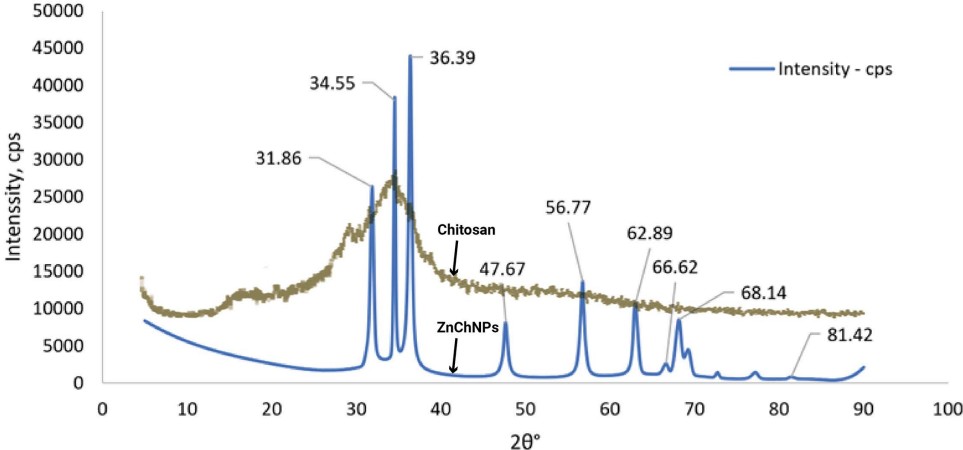

**Fig 6. X-ray diffraction (XRD) pattern of ZnChNPs.** X-ray diffraction (XRD) pattern of ZnChNPs, showing characteristic diffraction peaks plotted against 2θ (degree). The observed peaks indicate the crystalline nature of the synthesized nanoparticles and confirm the successful incorporation of zinc within the chitosan matrix.

### *In vivo* performance of ZnChNPs on bacterial leaf blight disease at net house conditions

The disease control group exhibited the highest disease severity, while the healthy control showed no symptoms. All treatment groups exhibited a remarkable reduction in lesion length, ranging from 36% to 73.54% at 14 days and 34.98% to 63.66% at 21 days after-inoculation compared to the disease control, indicating their effectiveness. Notably, treatments T5 (Bismerthiazol 0.15% + ZnChNP:dH$_2$O at 8:2) and T6 (Bismerthiazol 0.125% + ZnChNP:dH$_2$O at 8:2) achieved substantial reductions in lesion length. At 14 days, T5 produced the highest reduction of 68.28%, while T6 resulted in a 63.24% reduction. These results suggest a ratio-dependent effect, where a higher concentration of ZnChNPs corresponds to greater disease suppression. The efficacy of these treatments was sustained at 21 days, with reductions of 55.48% and 56.5%, respectively, indicating their potential for long-term disease management. Other treatments showed varying levels

**Table 1. FTIR spectra of ZnChNPs along with their possible functional group.**

| SI No. | Frequency (cm$^{-1}$) | Responsible Functional Group |
|---|---|---|
| 1. | 3394 | The O-H stretching group of alcohol |
| 2. | 2922 | The presence N-H stretching group of amine salt |
| 3. | 1626 | The C=N stretching group of imine/oxime and/or C=C stretching group of conjugated alkenes |
| 4. | 1486 | The N-O stretching group of a nitro compound |
| 5. | 1384 | The O-H bending group of carboxylic acid |
| 6. | 1261 | The C-O stretching group of alkyl aryl ether |
| 7. | 1107 | The S=O stretching group of sulfoxides |
| 8. | 1065 & 1007 | The presence of TPP crosslinker |
| 9. | 875 | The C=C bending group of alkenes |
| 10. | 467 | The stretching vibration of the Zn–O bond |

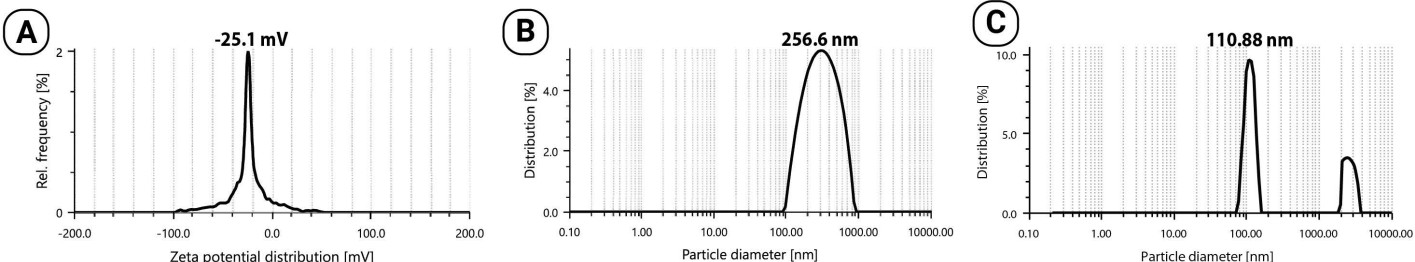

**Fig 7. Physical characterization of synthesized ZnChNPs. (A)** Zeta potential distribution curve showing a surface charge, **(B)** Dynamic Light Scattering (DLS) analysis illustrating the hydrodynamic diameter distribution, and **(C)** Particle size distribution.

of efficacy relative to the disease control, as illustrated in Fig 9. Additionally, ZnChNPs alone exhibited moderate effectiveness, reducing lesion length by 38.89% at 14 days and 29.82% at 21 days.

### *In vivo* effect of ZnChNPs on bacterial leaf blight under field conditions

This study evaluated the efficacy of combination therapy using zinc chitosan nanoparticles (ZnChNPs) and Bismerthiazol against bacterial leaf blight disease under field conditions. The performance of various Bismerthiazol and ZnChNPs ratios was compared with healthy and disease control groups. As expected, the healthy control group exhibited no lesions at both 14 and 21 days, indicating complete disease prevention, whereas the disease control group showed progressive BLB symptoms, with lesion lengths measuring 13.58 mm and 19.88 mm at 14 and 21 days, respectively. ZnChNPs alone demonstrated limited effectiveness, achieving 34.93% and 26.66% reductions in lesion length at 14 and 21 days after-inoculation, respectively. Notably, the T5 treatment (Bismerthiazol 0.15% + ZnChNP:dH$_2$O at 8:2) resulted in substantial reduction in lesion length at 66.14% and 61.27% at 14 and 21 days, respectively. Similarly, the T6 treatment (Bismerthiazol 0.125% + ZnChNP:dH$_2$O at 8:2) exhibited considerable efficacy, with reductions of 58.48% and 54.70% at the corresponding time points. Other treatments demonstrated variable effectiveness compared to the disease control, as illustrated in Fig 10. At 14 days, the T5 treatment even outperformed T4 (Bismerthiazol 0.2%) in disease suppression. A dose-dependent response was evident within the Bismerthiazol and ZnChNPs combinations, where higher doses generally correlated with greater reduction in disease severity. A representative image showing the effect of the T5 treatment alongside healthy and disease controls is presented in Fig 11.

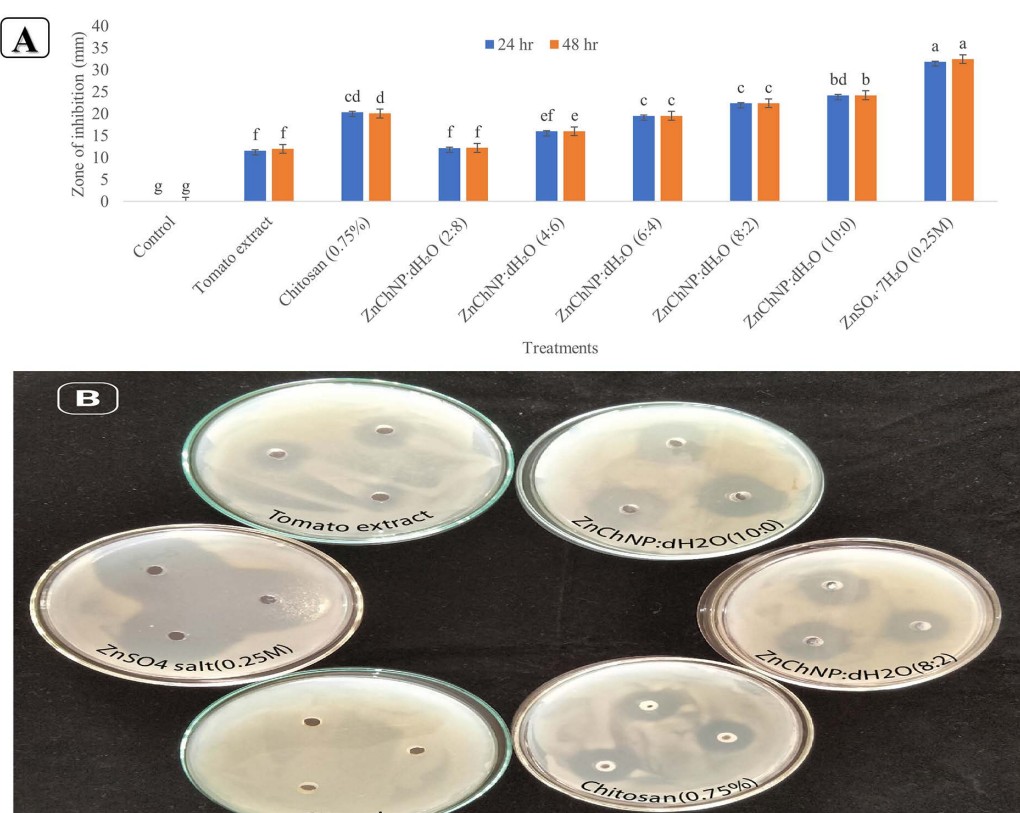

**Fig 8. Antimicrobial efficacy of ZnChNPs and precursors against Xanthomonas oryzae pv. oryzae. (A)** Comparative analysis of the Zone of Inhibition (ZoI) in mm across various treatments over a 21-days incubation at 28.9°C. Data represents the mean ± standard deviation. Distinct lowercase letters (a–g) above the bars indicate statistically significant differences between treatments (p < 0.05) according to the LSD post-hoc test (*p* = 0.05). Means followed by the same letter are not significantly different according to the LSD post-hoc test (*p* = 0.05). **(B)** Representative agar well diffusion plates showing the clear zones of inhibition for select treatments, including the negative control, tomato extract, chitosan (0.75%), and varying ratios of ZnChNPs.

## Discussion

Rice is a major source of nutrition for a significant proportion of the world's population [52–54]. The use of NPs is now a promising avenue of interest. This study demonstrated the green synthesis of ZnChNPs using a simple and environmentally friendly method. The UV-Visible absorption spectra of the synthesized ZnChNPs showed a characteristic peak at approximately 358 nm at room temperature. In agreement with our study, [55] reported a similar result for ZnCh Nanoparticles of different concentrations of ZnO in chitosan at a range of 360–348 nm. In contrast, [56] reported that the maximum peak for CSNPs was about 320–360 nm, whereas [30] found that the UV-vis spectra of zinc oxide nanoparticles showed a strong absorption band at 384, 380, and 386 nm for chamomile flower, olive leaves, and red tomato fruit, respectively.

The size and shape of nanoparticles play an important role in their antimicrobial activity against microbial pathogens, while smaller particle sizes can help nanoparticles easily enter the cell wall of microorganisms and increase the uptake of vehicles into the microbial cell [57]. In our study, the result of TEM as well as SEM, and FESEM images specified that the biosynthesized ZnChNPs have an uneven, diverse polymorphic shape. Similar results were previously reported by [55] and [49]. The TEM picture of these nanoparticles confirmed the effectiveness of the ionic gelation process in causing the agglomeration of Chitosan nanostructures when introducing ZnO nanoparticles. The result of this study aligned with

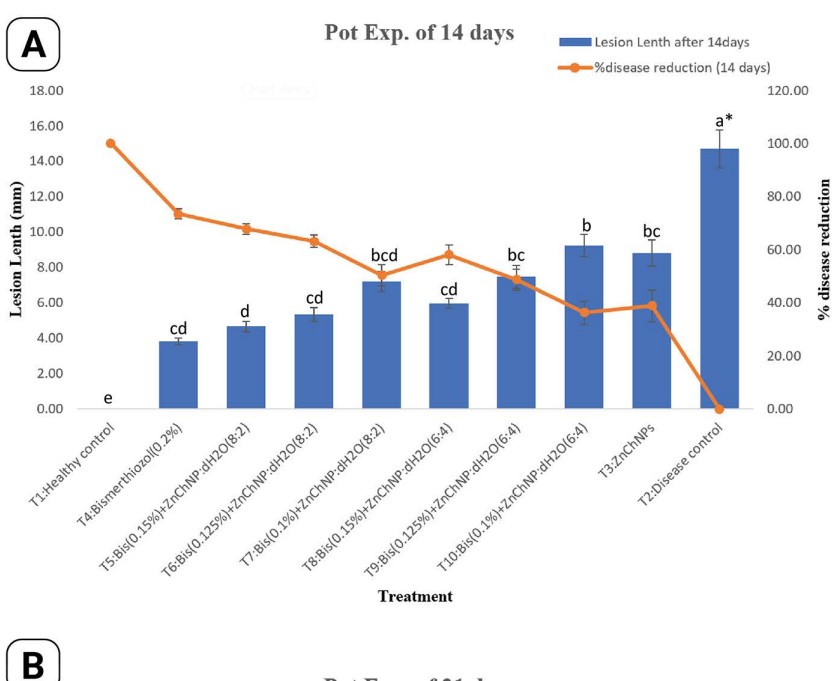

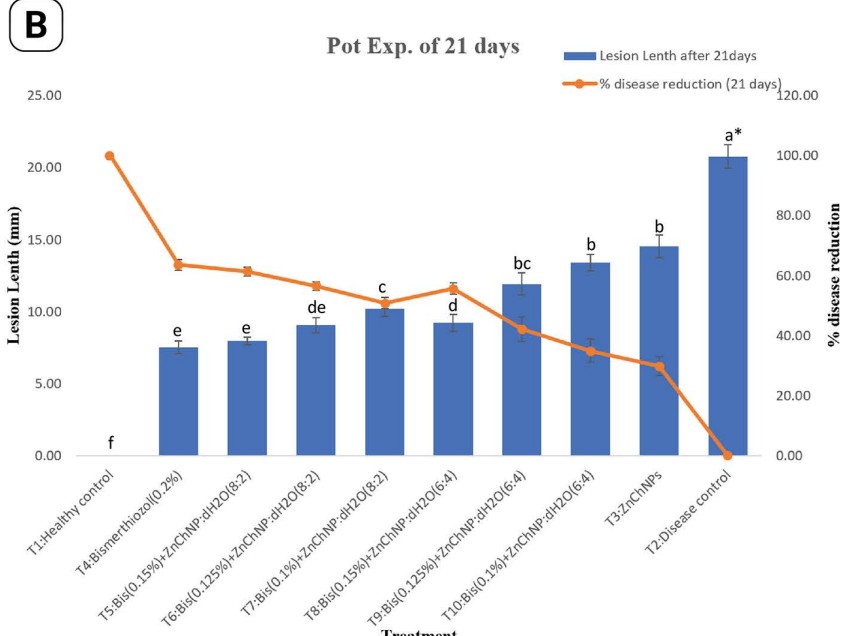

**Fig 9. Effect of different treatments against BLB disease of rice under nethouse conditions. (A)** Lesion length (mm) and percentage disease reduction at 14 days after inoculation (DAI). **(B)** Lesion length (mm) and percentage disease reduction at 21 DAI. Blue bars represent mean lesion length, and the orange line represents percentage disease reduction. Here, T1 = healthy control (non-inoculated), T2 = disease control (*Xoo*-inoculated, untreated), T3 = ZnChNP alone, T4 = Bismerthiazol (0.2%), T5 = Bismerthiazol (0.15%) + ZnChNP + $H_2O_2$ (8:2), T6 = Bismerthiazol (0.125%) + ZnChNP + $H_2O_2$ (8:2), T7 = Bismerthiazol (0.10%) + ZnChNP + $H_2O_2$ (8:2), T8 = Bismerthiazol (0.15%) + ZnChNP + $H_2O_2$ (6:4), T9 = Bismerthiazol (0.125%) + ZnChNP + $H_2O_2$ (6:4), and T10 = Bismerthiazol (0.10%) + ZnChNP + $H_2O_2$ (6:4). Error bars indicate standard deviation (mean ± SD). Different lowercase letters above bars indicate statistically significant differences among treatments according to Least Significant Difference (LSD) test ($p \le 0.05$).

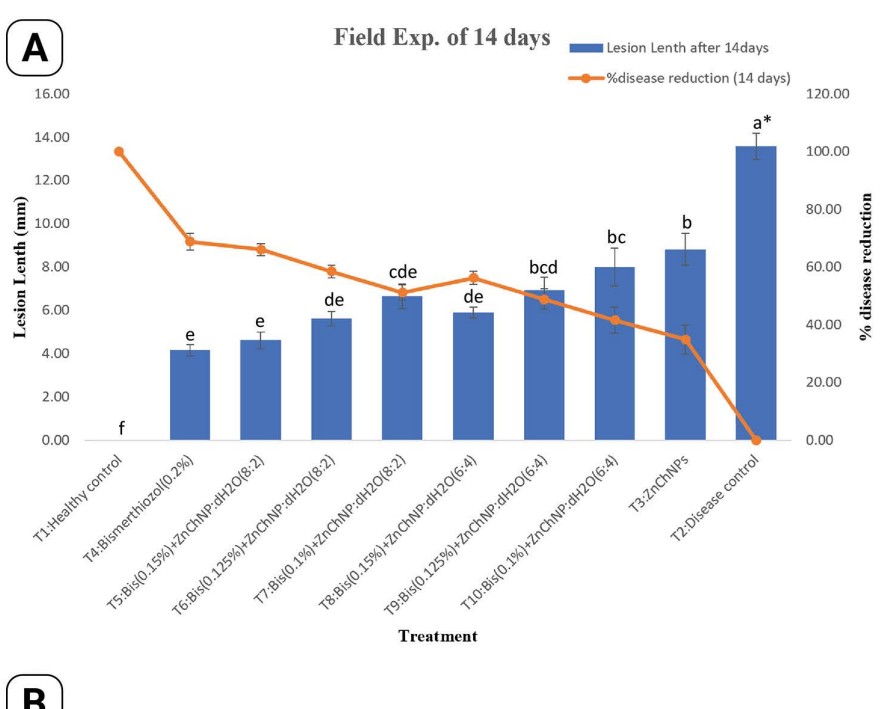

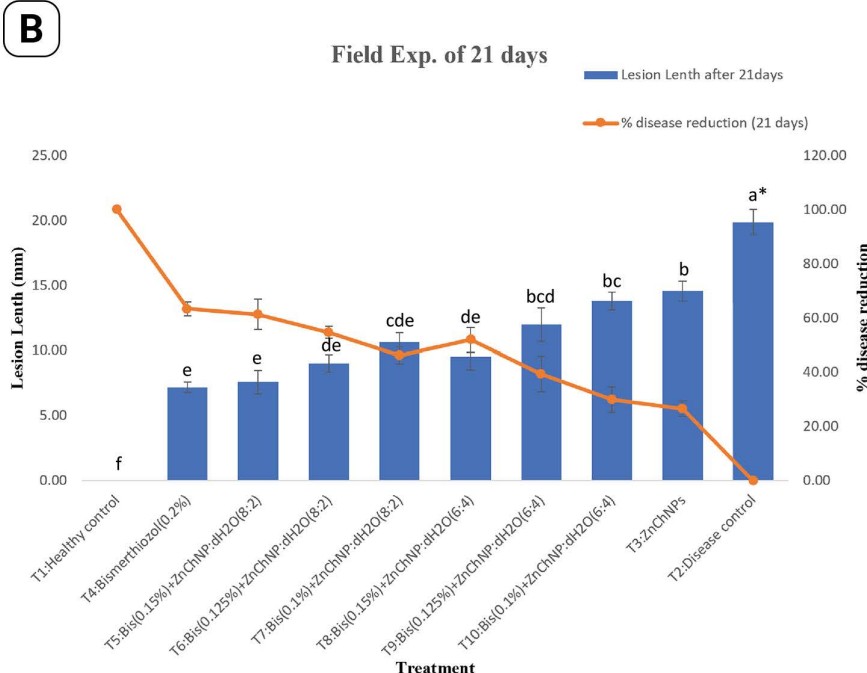

**Fig 10. Effect of different treatments against BLB disease of rice under field conditions. (A)** Lesion length (mm) and percentage disease reduction at 14 days after inoculation (DAI). **(B)** Lesion length (mm) and percentage disease reduction at 21 DAI. Blue bars represent mean lesion length, and the orange line represents percentage disease reduction. Here, T1 = healthy control (non-inoculated), T2 = disease control (*Xoo*-inoculated, untreated), T3 = ZnChNP alone, T4 = Bismerthiazol (0.2%), T5 = Bismerthiazol (0.15%) + ZnChNP + $H_2O_2$ (8:2), T6 = Bismerthiazol (0.125%) + ZnChNP + $H_2O_2$ (8:2), T7 = Bismerthiazol (0.10%) + ZnChNP + $H_2O_2$ (8:2), T8 = Bismerthiazol (0.15%) + ZnChNP + $H_2O_2$ (6:4), T9 = Bismerthiazol (0.125%) + ZnChNP + $H_2O_2$ (6:4), and T10 = Bismerthiazol (0.10%) + ZnChNP + $H_2O_2$ (6:4). Error bars indicate standard deviation (mean ± SD). Different lowercase letters above bars indicate statistically significant differences among treatments according to Least Significant Difference (LSD) test ($p ≤ 0.05$).

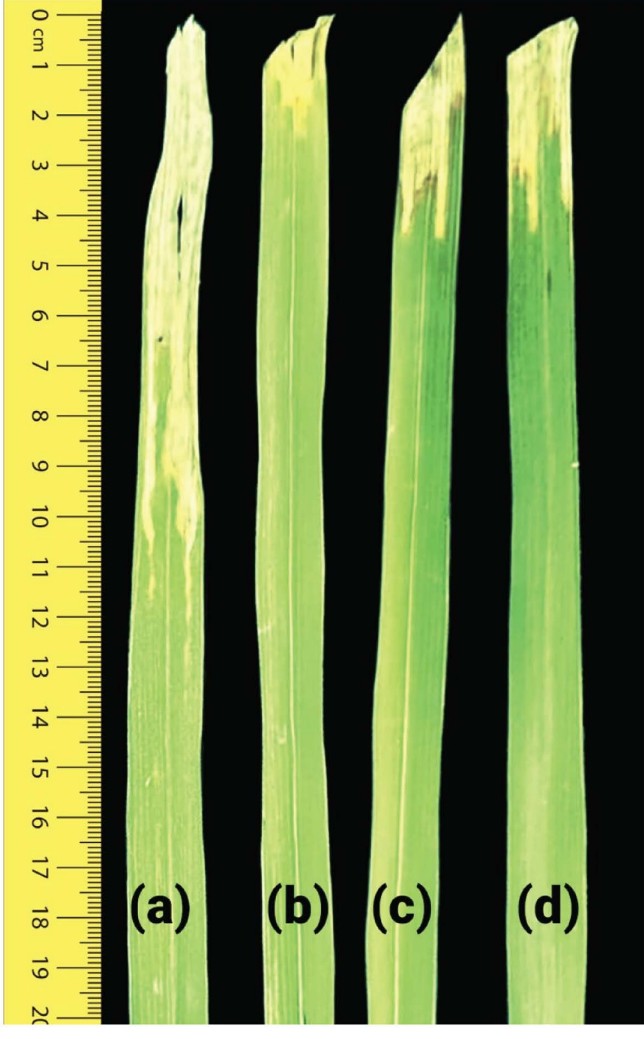

**Fig 11. Phenotypic response of rice plants to *Xanthomonas oryzae* pv. *oryzae* following ZnChNPs treatment. (A)** Untreated control plants showing typical disease symptoms, including leaf blight lesions and chlorosis. **(B–D)** Plants treated with ZnChNPs exhibiting reduced symptom severity, indicating enhanced disease resistance.

the observed potential of the ZnO NPs (negative charge) and CS/TPP NPs (positive charge [49] The SEM picture of the ZnChNPs displays spherical forms within a high-quality gelling system. Zinc di-cation species interact with the amino and hydroxyl groups on the chitosan scaffold, leading to an increase in the viscosity of the polymeric solution and the organization of the CS/TPP network [49,58]. The elemental maps show a consistent distribution pattern. The weight percentage of Zn on the surface was 29.46% greater than the measured value. The elemental composition of the biosynthesized ZnChNPs of this study is quite similar to the report of [30,31,49].

The stability of the green-synthesized NPs may be attributed, at least in part, to the presence of capping proteins, which could also confer additional antimicrobial properties. Indeed, the results of the FTIR spectrum provide an interpretation of the correlation between the absorption bands and the chemical compounds [59], which makes it possible for us to understand which biomolecules are involved in the increased antibacterial activity of nanoparticles. The FTIR study's data of ZnChNPs showed multiple absorption peaks at 3394, 2922, 1626, 1486, 1384, 1261, 1107, 1065, 1007, 875, and

467 cm$^{-1}$. In agreement with previous studies, the absorption peaks of ZnChNPs at 3394 and 2922 cm$^{-1}$ represented the elongation vibrations of the O-H stretching group of alcohol and the presence of the N-H stretching group of amine salt [31,49], respectively, while the absorption peak at 1626 cm$^{-1}$ was attributed to the C=N stretching group of imine/oxime and/or C=C stretching group of conjugated alkenes [31]. The absorption peaks observed at 1486 cm$^{-1}$ and 1384 cm$^{-1}$ correspond to the N–O stretching vibration of nitro compounds and the O–H bending vibration of carboxylic acid groups, respectively. The peaks at 1261 and 1107 cm$^{-1}$ revealed a C-O stretching group of alkyl aryl ether and an S=O stretching group of sulfoxide, respectively [31]. The bands detected at 1,065 and 1,008 cm$^{-1}$ indicate the presence of TPP crosslinker [49,60,61], while absorption peaks detected at 875 cm$^{-1}$ were attributed to the C=C bending group of alkenes [31]. As for the spectrum of ZnO, sharp bands observed at 467 cm$^{-1}$ were ascribed to the stretching vibration of the Zn–O bond [49,54,62]. In agreement with the result of this study, previous studies reported very similar FTIR peaks for the nanoparticles [31,49,55,63].

The XRD patterns of this experiment of ZnChNPs provided peaks at 2θ = 31.86°, 34.55°, 36.39°, 47.67°, 56.77°, 62.89°, 66.62°, 68.14, and 81.42° which corroborated the findings of previous studies. In the previous study, [49] showed peaks at 2θ 34.36°, 36.37°, 46.46°, 60.61°, and 66.06°, attributed to (002), (101), (102), (103), and (112) reflections of ZnO with hexagonal wurtzite structure [64]. [31] reported showed peaks at 2θ = 31.74°, 34.40°, 36.22°, 47.51°, 56.65°, 62.83°, and 69.05°, which were assigned to (100, 002, 101, 102, 110, 103, and 112) and observed at 2θ = 44.31°, 64.45°, and 81.77°, which were assigned to (200), (220), and (311) reflections for Chitosan NPs. In the previous study, [33] reported similar XRD diffraction peaks at 2θ = 31.8°, 34.5°, 36.3°, 47.6°, 56.6°, 62.9°, 68.0°, 72.6°, 77.0°, 81.4°, and 89.7° of ZnONPs synthesized by using *Matricaria chamomilla* plant extracts. The diffraction peaks were distinctly attributed to the hexagonal crystalline structure of ZnO nanoparticles. The narrow diffraction peaks of ZnO observed in the XRD pattern indicate that the samples were well crystallized [65] Click or tap here to enter text..

Based on dynamic light scattering (DLS) analysis, our findings revealed that the ZnChNPs possessed an average hydrodynamic size of 256.2 nm. This value is slightly larger than the particle size observed in transmission electron microscopy (TEM) micrographs. The larger particle sizes observed by DLS compared to SEM and TEM are attributed to the measurement of hydrodynamic diameter in aqueous suspension, which includes the solvation layer and surface-bound chitosan. In addition, partial aggregation and interparticle interactions in solution may contribute to the increased size detected by DLS, whereas SEM and TEM reflect the physical core size of dried nanoparticles. This discrepancy is attributed to the DLS method measuring the hydrodynamic radius, which includes the nanoparticle core along with any surrounding solvent molecules and surface ligands, unlike TEM which measures the dry core. This is presumably attributed to the high degree of complexation and stabilizing effect of phytochemicals toward Zn (II) ions at a relatively high concentration [49]. Upon exposure to an acidic chitosan solution containing TPP as a crosslinking agent, the nano-oxides and their precursors (Zn²⁺, SO₄²⁻) facilitated the formation of larger particles, ranging in size from 82.3 nm to 861.5 nm. The hydrodynamic size of ZnChNP was significantly increased when the CS/TPP mixture was added. The average particle diameter was determined to be 256.2 nm, with a PDI of 0.25. These results are consistent with those reported in prior studies, confirming the reproducibility of the formulation characteristics [49]. These are more often considered acceptable for polymeric nanomaterials [66]. The free ZnONPs exhibited a negative surface charge, whereas the ZnChNPs displayed a positive charge attributable to the ionized amino groups of chitosan forming the shell of the chitosan/tripolyphosphate (CS/TPP) complex. These nanoparticles were formed through the interaction between chitosan and TPP, resulting in CS-TPP complexes that were further crosslinked by excess TPP to create ordered colloidal structures. The ZP value of the ZnChNPs was lower due to the electrostatic interactions between ZnO NPs (−) → ← (+) CS/TPP NPs in the mixed system [49]. The formulation's stability was improved due to the large negative zeta potential value (−25.1 mV at pH 12.0).

The result of the current experiment demonstrated that the synthesized ZnChNPs exhibited potent antibacterial activity against the BLB pathogen of rice, *Xoo*. These findings are consistent with several previous reports [37,48]. Since *Xoo*

growth in agar medium stabilizes beyond 72 hours, only 24–48 hour measurements were considered for antibacterial evaluation, which is scientifically accepted for diffusion-based assays. Measurements beyond this period were excluded as they do not reflect active nanoparticle-mediated inhibition. However, the biosynthesized ZnO and Ag nanoparticles showed promising antibacterial activity against rice pathogen *Xoo* [47,48,66–68]. In contrast, ZnChNPs demonstrate a higher safety profile compared to metallic nanoparticles like silver or copper, as they do not contain heavy metal ions known to potentially disrupt plant development and soil microbial communities. In agreement with the result of this study, Abdallah et. al [31] showed the antibacterial activity of chitosan and ZnO nanoparticles against *Xoo*, and [37] reported a similar observation. However, this study suggests that ZnChNPs hold significant potential for application as effective antibacterial agents against bacterial pathogens in rice.

The *in vivo* studies in net house conditions as well as field conditions further confirmed the efficacy of ZnChNPs combined with Bismerthiazol in controlling BLB disease. This study depicts the potential of combining ZnChNPs with Bismerthiazol for BLB disease management. The T5: Bis (0.15%) +ZnChNP: dH2O (8:2) reduced 68.28% of the lesion length of BLB at 14 days after inoculation and 55.48% after 21 days compared to the disease control. Although the T5 treatment demonstrated a great effect in reducing lesion length compared to ZnChNPs alone at both 14 and 21 days, ZnChNPs-treated rice plants showed significantly reduced lesion areas compared to untreated control plants. This indicates that ZnChNPs possess inherent anti-lesion properties, even without the addition of Bismerthiazol. While all treatments offered protection, further investigation is needed to determine the optimal dosage and combination for long-term prevention, especially considering the limited data timeframe and lack of mechanistic insights. This observation suggests that ZnChNPs formulations may offer a dual advantage: potentially greater effectiveness at the outset (14 days) and more sustained efficacy over a longer period (21 days) compared to T3: Bismerthizol(0.2%) treatment. The biosynthesized ZnChNPs effectively reduced lesion length and damage caused by BLB on rice plants in both pot and field experiments, aligning with the results of the *in vitro* antibacterial activity. In agreement with our study, previous studies observed that AgNPs [67] and $Fe_3O_4$ and $TiO_2$ NPs [68] were able to significantly restrict the development of BLB lesion length as compared to the non-treated control in a pot experiment. The disease control potential of biosynthesized ZnChNPs, in the current study, T5: Bis (0.15%)+ZnChNP:dH2O (8:2) treatment with a 67.51% reduction in lesion length of the BLB was comparable to [69] who reported 63%−92% BLB disease inhibition by the application of various antibiotics and fungicides. Another study by [67] showed that pots treated with AgNP (15 µg/mL) *in vivo* in a greenhouse showed disease severity of 26.6% and disease decrease over control of 49.2%, at a much lower NP concentration than earlier reported studies from [66]. In another study, [70] concluded that foliar application of copper nanoparticles resulted in 90% and 15% reduction of BLB of pomegranate at early and mature disease stages, respectively, under controlled conditions, whereas 20% disease inhibition was measured under field conditions. [31] showed that the damage of zinc oxide and chitosan nanoparticles to the bacterial cells can cause cell death due to apoptosis, generation of ROS, reduction in biofilm formation and swimming, destruction or disintegration of the cell walls, and leakage of the intracellular contents [71–73]. In addition, many studies have reported that the primary mechanism for the inhibitory effect may be mainly due to the interaction between the positive charges of nanoparticles and the negatively charged compounds in the bacterial cell wall, which causes the penetration of nanoparticles into the bacterial cells, increasing membrane permeability, and intracellular flow [71,74–76]. Therefore, the antibacterial activity of the nanoparticles against *Xoo* may be partly due to their ability to damage the bacterial cell wall, which is essential for the bacteria's survival. [77] found that chitosan-Ag nanocomposites were able to generate ROS in *E. coli* cells. The formation of ROS promotes oxidative stress in the cells, which may in turn induce cell damage, causing cell lysis or distortion of bacterial membranes, resulting in the leakage of DNA and protein, and even bacterial death [78]. Indeed, some studies have indicated that the antibacterial activity of ZnChNPs could be due to the generation of ROS on oxide surfaces, which induced significant morphological changes and outflow in bacterial cells [79,80]. On the other hand, rotenone has been regarded as a scavenger of the electron transport chains in mitochondria, which is an essential site in the generation of ROS [81]. He attributes the significant reduction in ROS levels to the effect of rotenone.

Notably, the antibacterial efficacy of ZnChNPs has been reported to be influenced by their size and morphology, which are likely correlated with their specific modes of action into Nevertheless, five main mechanisms of antibacterial activities have been proposed, including (a) the production of reactive oxygen species (ROS), (b) the Interaction between ZnChNPs and bacterial surface, (c) photocatalytic action, (d) apoptosis, and (e) inhibition of biofilm formation and motility swimming [31,81].

The successful synthesis and characterization of ZnChNPs, coupled with their demonstrated *in vitro* and *in vivo* efficacy, suggest their significant potential as eco-friendly biopesticides for sustainable agriculture. ZnChNPs offer several advantages over conventional pesticides, including biodegradability, low toxicity, and broad-spectrum antibacterial activity. Furthermore, ZnChNPs can be synthesized through a simple and cheap method, rendering them suitable for large-scale production. Based on the promising findings of this study, we recommend further investigation into the potential applications of ZnChNPs in agriculture. ZnChNPs not only have an impact on BLB lesion length reduction, but they are also involved in plant growth under normal conditions, and plant stress tolerance mechanisms. Under chilling stress, supplementation with ZnONPs positively impacted the plant height, root length, and dry biomass of rice plants compared to the controls [5].

## Conclusion

Bacterial leaf blight is a major rice disease and require sustainable management strategy The results of *in vitro* and *in vivo* evaluations of this study established the effectiveness of ZnChNPs against BLB disease. Under *in vivo* conditions, ZnChNPs:dH$_2$O at 8:2 and 10:0 ratios reduced BLB lesion lengths by 36–73.54% at 14 days and 34.98–63.66% at 21 days. The combined Bismerthiazol (0.15%) + ZnChNPs:dH$_2$O (8:2) treatment achieved 68.28% and 55.48% lesion reduction in net house conditions, with field applications showing 66.14% and 61.27% reductions at 14 and 21 days, respectively. These results highlight the potential of ZnChNPs as an eco-friendly and effective nanotechnology-based alternative to conventional pesticides for controlling rice BLB disease while minimizing environmental risks. Further exploration of ZnChNPs in agriculture is recommended, focusing on optimizing synthesis for consistent nanoparticle properties, assessing long-term soil and environmental impacts, and conducting a detailed *in vitro* mechanistic study as a necessary next step for future research.

## Supporting information

**S1 Table. Weather data during the growing season (July–November 2023).** This file contains the environmental data cited in the main text.
(DOCX)

## Acknowledgments

The authors thank the Bangladesh Rice Research Institute (BRRI), Gazipur, Bangladesh, for providing research facilities.

## Author contributions

**Conceptualization:** Mohammad Abdul Latif.

**Data curation:** Md. Omar Kayess, A. L. Nayeem.

**Formal analysis:** Lutfur Rahman.

**Investigation:** Lutfur Rahman, Rumana Akter, Rakibul Hasan, A. K. M. Sahfiqul Islam, Sheikh Arafat Islam Nihad.

**Methodology:** Mohammad Abdul Latif.

**Project administration:** Mohammad Abdul Latif.

**Software:** Lutfur Rahman.

**Supervision:** A. K. M. Mohiuddin, Mohammad Abdul Latif.

**Validation:** Md. Omar Kayess.

**Visualization:** Md. Omar Kayess.

**Writing – original draft:** Lutfur Rahman.

**Writing – review & editing:** Md. Omar Kayess.

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
