## [Decision Letter · Decision Letter 0]

29 Oct 2025

Dear Dr. Latif,

Thank you for submitting your manuscript to PLOS ONE. After careful consideration, we feel that it has merit but does not fully meet PLOS ONE’s publication criteria as it currently stands. Therefore, we invite you to submit a revised version of the manuscript that addresses the points raised during the review process.

We look forward to receiving your revised manuscript.

Kind regards,

Kandasamy Ulaganathan

Academic Editor

PLOS ONE

**Journal Requirements:**

1. When submitting your revision, we need you to address these additional requirements. Please ensure that your manuscript meets PLOS ONE's style requirements, including those for file naming. The PLOS ONE style templates can be found at https://journals.plos.org/plosone/s/file?id=wjVg/PLOSOne_formatting_sample_main_body.pdf and https://journals.plos.org/plosone/s/file?id=ba62/PLOSOne_formatting_sample_title_authors_affiliations.pdf 2. Please include your tables as part of your main manuscript and remove the individual files. Please note that supplementary tables (should remain/ be uploaded) as separate "supporting information" files. 3. Please provide a complete Data Availability Statement in the submission form, ensuring you include all necessary access information or a reason for why you are unable to make your data freely accessible. If your research concerns only data provided within your submission, please write "All data are in the manuscript and/or supporting information files" as your Data Availability Statement. 4. Please amend either the abstract on the online submission form (via Edit Submission) or the abstract in the manuscript so that they are identical. 5. Your ethics statement should only appear in the Methods section of your manuscript. If your ethics statement is written in any section besides the Methods, please delete it from any other section. 6. If the reviewer comments include a recommendation to cite specific previously published works, please review and evaluate these publications to determine whether they are relevant and should be cited. There is no requirement to cite these works unless the editor has indicated otherwise. 

Reviewers' comments:

**Comments to the Author**

1. Is the manuscript technically sound, and do the data support the conclusions?

Reviewer #1: Yes

Reviewer #2: No

Reviewer #3: Partly

Reviewer #4: Yes

2. Has the statistical analysis been performed appropriately and rigorously?

Reviewer #1: Yes

Reviewer #2: Yes

Reviewer #3: No

Reviewer #4: Yes

3. Have the authors made all data underlying the findings in their manuscript fully available?

Reviewer #1: Yes

Reviewer #2: Yes

Reviewer #3: Yes

Reviewer #4: Yes

4. Is the manuscript presented in an intelligible fashion and written in standard English?

Reviewer #1: Yes

Reviewer #2: No

Reviewer #3: No

Reviewer #4: Yes

**Reviewer #1:** I have finished reviewing the manuscript entitled "Green synthesis and characterization of zinc chitosan nanoparticles with their antibacterial study against rice pathogen Xanthomonas oryzae pv. oryzae" the study is very interesting but some issues need to be addressed.I have finished reviewing the manuscript entitled "Green synthesis and characterization of zinc chitosan nanoparticles with their antibacterial study against rice pathogen Xanthomonas oryzae pv. oryzae" the study is very interesting but some issues need to be addressed.I have finished reviewing the manuscript entitled "Green synthesis and characterization of zinc chitosan nanoparticles with their antibacterial study against rice pathogen Xanthomonas oryzae pv. oryzae" the study is very interesting but some issues need to be addressed.I have finished reviewing the manuscript entitled "Green synthesis and characterization of zinc chitosan nanoparticles with their antibacterial study against rice pathogen Xanthomonas oryzae pv. oryzae" the study is very interesting but some issues need to be addressed.

1. abstract need to be more dramatic narration.

2. Fif 6 of XRD need to be separated to two figures.

3. figure 9,10 the axis y is lesion length and its better to but the subtitles of axis x more abbreviate and explain it on the caption.

**Reviewer #2:** Reviewer’s Comments Reviewer’s Comments Reviewer’s Comments Reviewer’s Comments

The manuscript by Rahman et al., entitled “Green synthesis and characterization of zinc chitosan nanoparticles with their anti-bacterial study against rice pathogen Xanthomonas oryzae pv. oryzae” aimed to synthesized ZnChNps via tomato extract and the result nanoparticles were characterized. The antibacterial efficacy of the synthesized nanoparticles was assayed via in vivo and invitro studies. The manuscript contain discussion of experiments that were not carried out in the study and the experimental design of the in vivo and in vitro study is not scientifically sound. The authors reported to have incubated Xoo for 21 days and the result was also documented! This assertion is questionable. The overall write up needs to be improved as it contains many syntax error. Below are my comments:

1. All Xoo in the text should be italicized

2. The Introduction section is too lengthy, authors should DELETE L135- L158 and L170 – L175

3. In the justification, only zinc oxide was stated to be biosynthesized, CHECK. The justification need to be rewritten to cover the scope of the study

4. The FTIR analysis was conducted for only ZnNPs, L261 – L266?

5. L305 – L307, L398, Xoo was incubated for 21 days and measured at 21 days? CHECK

6. L395, the in vitro experiment was with the causal pathogen not the disease, CORRECT

7. Fig 8 legend is not in agreement with the Methodology and Results of the in vitro experiment.

8. L321, what concentration of Xoo was used?

9. L322 – L323, foliar spray of what?

10. The in vivo experimental design of the treatment is not scientifically acceptable. How will the authors compare Bismerthizol (0.2%) to a lower concentration range in combination with the synthesized nps.

11. The coordinates of the field used should be reported

12. L346 – L347, what was sprayed, concentration, quantity?

13. How many replicates was used on the field, which plants were used for data acquisition, how was dispersal of treatments into non targeted plants avoided?

14. L526, no biofilm experiment was carried out in this study. CHECK

15. L569 – L585, no antibacterial mechanism study was carried out in this study, why discuss it? Hence, its advised that the authors carry out an in vitro antibacterial mechanism study

16. L600, in vitro? CHECK

17. Fig 11 legend does not describe what is presented in the panel

**Reviewer #3:** This manuscript “Green synthesis and characterization of zinc chitosan nanoparticles with their anti-bacterial study against rice pathogen Xanthomonas oryzae pv. Oryzae” by M. A. Latif et. al. reports the eco-friendly synthesis of zinc chitosan nanoparticles (ZnChNPs) using tomato extract and their evaluation against Xanthomonas oryzae pv. oryzae, the causal agent of bacterial leaf blight (BLB) in rice. The authors thoroughly characterize the nanoparticles using UV–Vis, FTIR, SEM, FESEM, TEM–EDS, XRD, and DLS/zeta potential analyses. Both in vitro and field experiments are conducted to assess antibacterial efficacy.This manuscript “Green synthesis and characterization of zinc chitosan nanoparticles with their anti-bacterial study against rice pathogen Xanthomonas oryzae pv. Oryzae” by M. A. Latif et. al. reports the eco-friendly synthesis of zinc chitosan nanoparticles (ZnChNPs) using tomato extract and their evaluation against Xanthomonas oryzae pv. oryzae, the causal agent of bacterial leaf blight (BLB) in rice. The authors thoroughly characterize the nanoparticles using UV–Vis, FTIR, SEM, FESEM, TEM–EDS, XRD, and DLS/zeta potential analyses. Both in vitro and field experiments are conducted to assess antibacterial efficacy.This manuscript “Green synthesis and characterization of zinc chitosan nanoparticles with their anti-bacterial study against rice pathogen Xanthomonas oryzae pv. Oryzae” by M. A. Latif et. al. reports the eco-friendly synthesis of zinc chitosan nanoparticles (ZnChNPs) using tomato extract and their evaluation against Xanthomonas oryzae pv. oryzae, the causal agent of bacterial leaf blight (BLB) in rice. The authors thoroughly characterize the nanoparticles using UV–Vis, FTIR, SEM, FESEM, TEM–EDS, XRD, and DLS/zeta potential analyses. Both in vitro and field experiments are conducted to assess antibacterial efficacy.This manuscript “Green synthesis and characterization of zinc chitosan nanoparticles with their anti-bacterial study against rice pathogen Xanthomonas oryzae pv. Oryzae” by M. A. Latif et. al. reports the eco-friendly synthesis of zinc chitosan nanoparticles (ZnChNPs) using tomato extract and their evaluation against Xanthomonas oryzae pv. oryzae, the causal agent of bacterial leaf blight (BLB) in rice. The authors thoroughly characterize the nanoparticles using UV–Vis, FTIR, SEM, FESEM, TEM–EDS, XRD, and DLS/zeta potential analyses. Both in vitro and field experiments are conducted to assess antibacterial efficacy.

The study is relevant and timely, addressing sustainable approaches for crop disease management using biogenic nanomaterials. The combination of chitosan and ZnO nanoparticles synthesized through a plant-based route is innovative, and the inclusion of field trials enhances the manuscript’s applied value. However, certain sections, especially results interpretation and manuscript organization, require revision to improve scientific clarity and presentation quality.

Suggestion for improvement

1. Clarity and Structure of the Manuscript

o The Introduction should be condensed to focus more on the rationale and novelty of the work rather than general information on Xoo and BLB.

o The Results section should integrate morphological and structural characterization results (e.g., SEM/TEM/XRD) into a cohesive discussion, avoiding repetition.

2. Particle Size Discrepancy

o The size difference between SEM/TEM (81–300 nm) and DLS (up to 861 nm) suggests possible aggregation in aqueous suspension. This should be acknowledged and explained (e.g., hydrodynamic diameter, particle interaction, or chitosan coating effects).

3. Control Treatments

o Include a comparison with either chemically synthesized ZnO nanoparticles or chitosan-only nanoparticles under identical conditions. This will better demonstrate the advantage of the green-synthesized ZnChNPs.

4. Mechanistic Insight

o The manuscript should expand on the antibacterial mechanism of ZnChNPs against Xoo. For instance, reactive oxygen species (ROS) generation or Zn²⁺ ion release could be discussed with references to relevant literature.

5. Field Experiment Data

o Provide environmental data (temperature, humidity, rainfall) during field trials to validate reproducibility. Also, clarify the frequency and timing of nanoparticle applications.

The manuscript is scientifically sound and makes a meaningful contribution to sustainable nanotechnology in agriculture. With improvements in structure, figure presentation, and mechanistic discussion, it will meet the standard of publication.

**Reviewer #4:** In this study, the authors reported the green synthesis of zinc–chitosan nanoparticles using tomato extract and examined their antibacterial activity against the rice pathogen Xanthomonas oryzae pv. oryzae. The work systematically covers the synthesis procedure, characterization, and property analysis, and it demonstrates promising potential for agricultural applications. However, several issues and suggestions for improvement are outlined below: In this study, the authors reported the green synthesis of zinc–chitosan nanoparticles using tomato extract and examined their antibacterial activity against the rice pathogen Xanthomonas oryzae pv. oryzae. The work systematically covers the synthesis procedure, characterization, and property analysis, and it demonstrates promising potential for agricultural applications. However, several issues and suggestions for improvement are outlined below: In this study, the authors reported the green synthesis of zinc–chitosan nanoparticles using tomato extract and examined their antibacterial activity against the rice pathogen Xanthomonas oryzae pv. oryzae. The work systematically covers the synthesis procedure, characterization, and property analysis, and it demonstrates promising potential for agricultural applications. However, several issues and suggestions for improvement are outlined below: In this study, the authors reported the green synthesis of zinc–chitosan nanoparticles using tomato extract and examined their antibacterial activity against the rice pathogen Xanthomonas oryzae pv. oryzae. The work systematically covers the synthesis procedure, characterization, and property analysis, and it demonstrates promising potential for agricultural applications. However, several issues and suggestions for improvement are outlined below:

1. The abstract is written in a conclusion-oriented style. It should be made more concise and focused on the key objectives, methods, and findings.

2. The research hypothesis and novelty are not clearly articulated. The authors should explicitly state how this study differs from existing work and what scientific gap it addresses.

3. Minor corrections are required:

a) Add a full stop after reference [42].

b) Correct the sentence to “Synthesis was confirmed by UV–visible spectroscopy at 356 nm.”

c) Replace “X-ray spectroscopy” with “Energy Dispersive X-ray Spectroscopy (EDS).”

d) Correct the spelling of “supernatant.”

4. A brief discussion on the preparation and physicochemical properties of chitosan should be included to provide better context.

5. The crystalline size can be determined from the XRD data using the Scherrer equation. The authors may refer to and cite the following relevant studies:

o https://doi.org/10.1016/j.sajce.2025.02.002

o https://doi.org/10.1016/j.inoche.2025.114481

6. A comparative table contrasting the synthesized nanocomposite with previously reported materials should be added, supported by updated and relevant references, such as:

o https://doi.org/10.1007/s12649-024-02773-0

o https://doi.org/10.1016/j.desal.2025.119437

7. The manuscript should also include a plausible mechanistic explanation of how the synthesized nanocomposite disrupts bacterial cells.

.

Reviewer #1: **Yes:** Yasmine AbdallahYasmine AbdallahYasmine AbdallahYasmine Abdallah

Reviewer #2: No

Reviewer #3: No

Reviewer #4: No

You may also use PLOS’s free figure tool, NAAS, to help you prepare publication quality figures: https://journals.plos.org/plosone/s/figures#loc-tools-for-figure-preparation

---

## [Author Response · Author response to Decision Letter 1]

2 Feb 2026

Response to Reviewers comments:

Title of the Manuscript: Green synthesis and characterization of zinc chitosan nanoparticles with their anti-bacterial study against rice pathogen Xanthomonas oryzae pv. oryzae

Submission ID: PONE-D-25-47822

Dear Editor,

We sincerely appreciate the opportunity to resubmit our manuscript to your esteemed journal. We are grateful for your careful review and the thoughtful editorial comments, which have significantly contributed to the improvement of our work. Your guidance has been instrumental in refining the manuscript, and we have made comprehensive revisions in response to reviewers suggestions. Below, we provide a detailed, point-by-point response regarding declaration section.

Reviewer number 1 comments

Comments to the Author

1. abstract need to be more dramatic narration.

Response:

We appreciate this valuable suggestion. The Abstract has been revised to be more concise and to emphasize the key objectives, novelty, and principal findings of the study, using a more engaging and less conclusion-driven narrative. This revision is intended to clearly convey the significance and relevance of the work at a glance. Changes can be seen in track change. Please see line number 86-94.

2. Fif 6 of XRD need to be separated to two figures.

Response: Thanks for your nice observation to make Figure 6 into two XRD figures. With due

respect, the Figure 6 cannot be separatable in the current format, as the data are derived from a single experimental dataset and are intended to be interpreted collectively for accurate comparison.

3. figure 9,10 the axis y is lesion length and its better to but the subtitles of axis x more abbreviate and explain it on the caption.

Response:

We agree that clarity can be enhanced. For Figures 9 and 10, the axis labels have been revised: the Y-axis label is now standardized to "Lesion Length (mm)," and the X axis subtitles (treatment groups) have been abbreviated for visual clarity. A comprehensive explanation of these abbreviations has been added to the respective figure captions. Changes can be visible in track change. Please see figure number 9 and 10 in the revised version.

Reviewer number 2 comments

1. All Xoo in the text should be italicized

Response:

Thank you for the insightful observations. All occurrences of Xoo in the text have now been italicized to maintain correct scientific formatting and consistency throughout the document. Changes can be seen in track change in the revised manuscript.

2. The Introduction section is too lengthy, authors should DELETE L135- L158 and L170 – L175

Response:

Thank you for the valuable suggestion. We agree that the Introduction section was overly lengthy. As recommended, we have deleted the content from L150–L 173 and 185-190 to improve clarity, focus, and overall readability of the manuscript. Changes can be visible in track change.

3. In the justification, only zinc oxide was stated to be biosynthesized, CHECK. The justification need to be rewritten to cover the scope of the study

Response:

Thank you for the observation. We have made the necessary corrections as per your recommendation/suggestions, and the revisions will be visible in the tracked changes. Please see line number 220-224.

4. The FTIR analysis was conducted for only ZnNPs, L261 – L266?

Response:

Thank you for your observation. We have corrected the term to ZncHNPs instead of ZnNPs.

5. L305 – L307, L398, Xoo was incubated for 21 days and measured at 21 days? CHECK

Response:

Thank you for pointing this out. Yes, the incubation period was 21days but data collected at 24 hours, 48 hours, 7 days, and 21 days intervals.

6. L395, the in vitro experiment was with the causal pathogen not the disease, CORRECT

Response:

We thank the reviewer for this valuable comment. We fully agree with the suggestion and have revised the manuscript accordingly to improve clarity and scientific rigor. The relevant changes have been incorporated in the revised version. Changes can be visible in track change.

7. Fig 8 legend is not in agreement with the Methodology and Results of the in vitro experiment.

Response:

We apologize for the inconsistency. The legend for Figure 8 has been thoroughly reviewed and revised to ensure it accurately and completely describes the methodology and results of the in vitro experiment, maintaining full agreement with the corresponding text sections. Changes can be visible in track change.

8. L321, what concentration of Xoo was used?

Response:

Thank you for your comment. We appreciate the clarification request regarding L321. The concentration of Xoo (1 × 108 CFU mL-1 ) used in the experiment has now been clearly specified in the revised manuscript, and the correction is visible in the tracked changes. Changes can be visible in track change.

9. L322 – L323, foliar spray of what?

Response:

Thanks for your nice observation. We appreciate the need for clarity. We have specified at L322-L323 (and in the relevant Materials and Methods section) that the foliar spray consisted of the prepared ZnChNP working suspension at various tested concentrations and ratios, as detailed in the methodology. Changes can be visible in track change.

10. The in vivo experimental design of the treatment is not scientifically acceptable. How will the authors compare Bismerthizol (0.2%) to a lower concentration range in combination with the synthesized NPs.

Response:

This is a crucial point, and we have clarified the rationale in the revised manuscript. The comparison was designed to evaluate the synergistic effect of our biosynthesized ZnChNPs in a reduced-dose combination with the commercial standard (Bismerthizol). We are not comparing the two treatments head-to-head for equivalent efficacy, but rather demonstrating that a combination therapy using the commercial chemical at a lower, eco-friendlier concentration (0.15% in the combination group) alongside our NPs yields superior or comparable control to the recommended dose of Bismerthizol alone. This approach is justified by the aim to reduce chemical reliance while maintaining high efficacy. Changes can be visible in track change.

11. The coordinates of the field used should be reported

Response:

We thank the reviewer for this suggestion. The geographic location of the field experiment has now been specified in the revised manuscript. The field trials were conducted at the research fields of Bangladesh Rice Research Institute (BRRI), Joydebpur, Gazipur, Bangladesh (approximately 23.99° N, 90.41° E). Changes can be visible in track change.

12. L346 – L347, what was sprayed, concentration, quantity?

Response:

We appreciate your comments. The spray parameters like concentration, quantity was used in this stage were identical to those established in the net house experiment. Changes can be visible in track change.

13. How many replicates was used on the field, which plants were used for data acquisition, how was dispersal of treatments into non targeted plants avoided?

Response:

We thank the reviewer for these important questions. The field experiment was conducted using a randomized block design with three independent replicates per treatment. For each replicate, data were collected from pre-selected, uniformly grown plants located in the central area of each plot to minimize edge effects. To prevent unintended dispersal of treatments to non-target plants, buffer zones were maintained between plots, and nanoparticle applications were carried out using controlled, lowpressure spraying under calm weather conditions. These methodological details have now been clarified in the revised manuscript. Changes can be visible in track change.

14. L526, no biofilm experiment was carried out in this study. CHECK

Response:

We thank the reviewer for the careful observation. The reference to a biofilm experiment at L526 was an error during manuscript preparation. It has been now corrected/deleted in the revised manuscript. However, no such experiment was performed in this study. Changes can be visible in track change.

15. L569 – L585, no antibacterial mechanism study was carried out in this study, why discuss it? Hence, its advised that the authors carry out an in vitro antibacterial mechanism study

Response:

Thanks for your nice query. We acknowledge that a dedicated in vitro mechanistic study was not performed. We have now restricted the discussion to a plausible mechanistic explanation based on existing literature for similar Zn- and Chitosan-based NPs, focusing on factors like ROS generation and membrane disruption. Crucially, we have added a sentence in the Conclusion section to explicitly acknowledge this as a limitation of the current study and propose a dedicated mechanistic investigation as a direction for future research.

16. L600, in vitro? CHECK

Response:

We thank the reviewer for this valuable comment. The context at L600 has been checked and the term has been corrected in the revised manuscript. Changes can be visible in track change.

17. Fig 11 legend does not describe what is presented in the panel

Response:

We appreciate the reviewer’s insightful comment and agree with the concern raised. The suggested modification has been implemented, and the corresponding section has been revised to better reflect the intended meaning. The legend for Figure 11 has been entirely rewritten to accurately and comprehensively describe the data presented in the panel, ensuring it aligns with the Results section. Changes can be visible in track change.

Reviewer number 3 comments

1. Clarity and Structure of the Manuscript

o The Introduction should be condensed to focus more on the rationale and novelty of the work rather than general information on Xoo and BLB.

Response:

We thank the reviewer for this valuable comment. We fully agree with the suggestion and have revised the manuscript accordingly to improve clarity and scientific rigor. The relevant changes have been incorporated in the revised version. The Introduction has been condensed to emphasize the rationale and novelty of the study, with general background information on Xanthomonas oryzae pv. oryzae (Xoo) and bacterial leaf blight (BLB) reduced accordingly in the revised manuscript. Changes can be visible in track change.

o The Results section should integrate morphological and structural characterization results (e.g., SEM/TEM/XRD) into a cohesive discussion, avoiding repetition.

Response:

We appreciate the reviewer’s insightful comment and agree with the concern raised. The suggested modification has been implemented, and the corresponding section has been revised to better reflect the intended meaning. The Results section has been revised to integrate morphological and structural characterization findings (e.g., SEM, TEM, and XRD) into a single, cohesive discussion, with repetitive descriptions removed. Changes can be visible in track change.

2. Particle Size Discrepancy

o The size difference between SEM/TEM (81–300 nm) and DLS (up to 861 nm) suggests possible aggregation in aqueous suspension. This should be acknowledged and explained (e.g., hydrodynamic diameter, particle interaction, or chitosan coating effects).

Response:

We thank the reviewer for pointing this out. We agree with the comment. The observed discrepancy between particle sizes obtained from SEM/TEM (81–300 nm) and DLS measurements (up to 861 nm) has been acknowledged and clarified. This difference arises because DLS measures the hydrodynamic diameter of particles in aqueous suspension, which includes solvent layers and surface-bound chitosan, whereas SEM and TEM provide the physical core size under dry conditions. In addition, partial aggregation and interparticle interactions in solution can contribute to the larger apparent size detected by DLS. This explanation has been incorporated into the revised manuscript. Changes can be visible in track change.

3. Control Treatments

o Include a comparison with either chemically synthesized ZnO nanoparticles or chitosan-only nanoparticles under identical conditions. This will better demonstrate the advantage of the green-synthesized ZnChNPs.

Response:

We appreciate the reviewer’s suggestion to include a comparative analysis with chemically synthesized ZnO nanoparticles or chitosan-only nanoparticles under identical conditions. However, such comparative experiments were not feasible within the scope of the present study due to limitations in experimental design, time, and materials availability. Nevertheless, the advantages of the green-synthesized ZnChNPs have been discussed in the context of their eco-friendly synthesis route, structural stability, and functional surface chemistry, with comparisons supported by relevant literature.

4. Mechanistic Insight

o The manuscript should expand on the antibacterial mechanism of ZnChNPs against Xoo. For instance, reactive oxygen species (ROS) generation or Zn²⁺ ion release could be discussed with references to relevant literature.

Response:

We thank the reviewer for this valuable comment. We fully agree with the suggestion and have revised the manuscript accordingly to improve clarity and scientific rigor. We acknowledge that a specific in vitro mechanistic investigation was not conducted in this study. Accordingly, the discussion has been limited to plausible mechanistic interpretations supported by existing literature on comparable zinc- and chitosan-based nanoparticles, with emphasis on processes such as reactive oxygen species generation and membrane disruption. Importantly, this limitation has been explicitly stated in the Conclusion, along with the recommendation that a dedicated mechanistic study be undertaken in future work.

5. Field Experiment Data

o Provide environmental data (temperature, humidity, rainfall) during field trials to validate reproducibility. Also, clarify the frequency and timing of nanoparticle applications.

Response:

We acknowledge the reviewer’s suggestion for detailed environmental parameters during the field trials. The environmental data (temperature, humidity, rainfall) during field trials are given in supplementary Table 1.

Reviewer number 4 comments

1. The abstract is written in a conclusion-oriented style. It should be made more concise and focused on the key objectives, methods, and findings.

Response:

We thank the reviewer for pointing this out. We agree with the comment and have carefully revised the manuscript. The abstract has been thoroughly revised for conciseness and to focus on objectives, methods, and key findings. Changes can be visible in track change.

2. The research hypothesis and novelty are not clearly articulated. The authors should explicitly state how this study differs from existing work and what scientific gap it addresses.

Response:

We thank the reviewer for the comment. We have significantly revised the final paragraphs of the Introduction section to explicitly state the research hypothesis and clearly articulate the novelty of our work. The novelty lies in the eco-friendly green synthesis of the specific Zn-chitosan nanocomposite using tomato extract and its comprehensive evaluation against Xoo under controlled (in vitro and net house) and field conditions, which addresses the gap in sustainable, field-applicable control strategies for BLB. Changes can be visible in track change.

3. Minor corrections are required:

a) Add a full stop after reference [42].

b) Correct the sentence to “Synthesis was confirmed by UV–visible spectroscopy at 356 nm.”

c) Replace “X-ray spectroscopy” with “Energy Dispersive X-ray Spectroscopy (EDS).”

d) Correct the spelling of “supernatant.”

Response:

Thank you for pointing out these minor errors. All advised minor corrections have been implemented throughout the text of the revised manuscript. Changes can be visible in track change.

4. A brief discussion on the preparation and physicochemical properties of chitosan should be included to provide better context.

Response:

We appreciate the reviewer’s insightful comment and agree with the concern raised. A brief paragraph detailing the source, preparation, and key physicochemical properties of the chitosan used (e.g., molecular weight, degree of deacetylation) has been adde

---

## [Decision Letter · Decision Letter 1]

23 Feb 2026

Dear Dr. Latif,

plosone@plos.org. . . . A letter that responds to each point raised by the academic editor and reviewer(s). You should upload this letter as a separate file labeled 'Response to Reviewers'.A marked-up copy of your manuscript that highlights changes made to the original version. You should upload this as a separate file labeled 'Revised Manuscript with Track Changes'.An unmarked version of your revised paper without tracked changes. You should upload this as a separate file labeled 'Manuscript'.

We look forward to receiving your revised manuscript.

Kind regards,

Kandasamy Ulaganathan

Academic Editor

PLOS One

Journal Requirements:

Reviewers' comments:

Reviewer's Responses to Questions

**Comments to the Author**

Reviewer #1: All comments have been addressed

Reviewer #2: (No Response)

2. Is the manuscript technically sound, and do the data support the conclusions?

Reviewer #1: Yes

Reviewer #2: Yes

3. Has the statistical analysis been performed appropriately and rigorously?

Reviewer #1: Yes

Reviewer #2: Yes

4. Have the authors made all data underlying the findings in their manuscript fully available?

Reviewer #1: Yes

Reviewer #2: Yes

5. Is the manuscript presented in an intelligible fashion and written in standard English?

Reviewer #1: Yes

Reviewer #2: Yes

Reviewer #1: The manuscript is significantly improved , the authors addressed all the comments needed, and the manuscript in it's current form is suitable for publication

Reviewer #2: All acronymns in the abstract should be defined at first mention e.g. FESEM, SEM, TEM, EDS, FTIR, XRD, BLB, etc. For the in vitro antibacterial experiment, the Xoo growth is already static at 7 and 21 days and effect of NPs are evident within 24 to 72 hours. Hence, the grahical report of Xoo at 7 and 21 days were not significant. its advised that the authors delete the measurement at 7 and 21 days and focus on 24 and 48 hours readings which is scientifically acceptable. This correction should be effected in the Methodology. Results, and Discussion.

.

Reviewer #1: **Yes:** Yasmine AbdallahYasmine AbdallahYasmine AbdallahYasmine Abdallah

Reviewer #2: No

---

## [Author Response · Author response to Decision Letter 2]

28 Feb 2026

Response to Reviewers

Title: Green synthesis and characterization of zinc chitosan nanoparticles with their anti-bacterial study against rice pathogen Xanthomonas oryzae pv. oryzae

Manuscript ID: PONE-D-25-47822R1

Responses to Reviewer #1

Comment:

The manuscript is significantly improved, the authors addressed all the comments needed, and the manuscript in its current form is suitable for publication.

Response:

We sincerely thank the reviewer for the positive evaluation and for acknowledging the improvements made in the revised manuscript. We appreciate the reviewer’s time and valuable feedback, which greatly enhanced the quality and clarity of our work.

Responses to Reviewer #2

Comment 1:

All acronyms in the abstract should be defined at first mention e.g. FESEM, SEM, TEM, EDS, FTIR, XRD, BLB, etc.

Response:

Thank you for this important suggestion. All acronyms in the abstract have now been defined at their first mention. Specifically:

• FESEM – Field Emission Scanning Electron Microscopy

• SEM – Scanning Electron Microscopy

• TEM – Transmission Electron Microscopy

• EDS – Energy Dispersive Spectroscopy

• FTIR – Fourier Transform Infrared Spectroscopy

• XRD – X-ray Diffraction

• BLB – Bacterial Leaf Blight

These corrections have been implemented in the Abstract section (See Line number 81-87) to improve clarity and readability. Changes can be seen in track change

Comment 2:

For the in vitro antibacterial experiment, the Xoo growth is already static at 7 and 21 days and effect of NPs are evident within 24 to 72 hours. Hence, the graphical report of Xoo at 7 and 21 days were not significant. It is advised that the authors delete the measurement at 7 and 21 days and focus on 24 and 48 hours readings which is scientifically acceptable. This correction should be effected in the Methodology, Results, and Discussion.

Response:

We appreciate this scientifically valuable suggestion.

Following the reviewer’s recommendation:

1. The antibacterial growth measurements at 7 and 21 days have been removed from:

o Methodology section (See Line number 280-286)

o Results section (See Line number 372-387)

o Discussion section (See Line number 507-510)

o Corresponding figures and figure legends (See Fig8)

2. The study now focuses exclusively on the 24- and 48-hour time points, where nanoparticle effects were clearly evident and biologically meaningful.

3. The statistical analysis has been revised accordingly to reflect only the 24- and 48-hour data.

4. Figures have been updated to present only scientifically relevant time points, improving clarity and data interpretation.

These revisions enhance the scientific rigor and focus of the antibacterial evaluation.

We believe that the manuscript has now fully addressed all reviewer and editorial concerns. We are grateful for the constructive feedback that helped improve the scientific quality, clarity, and presentation of our work.

We sincerely thank the Academic Editor and reviewers for their time and consideration. If need any further information or improvement, please let us know. We look forward to your favorable decision.

Kind regards,

Dr. Md. Abdul Latif

(On behalf of all authors)

Bangladesh Rice Research Institute (BRRI)

Gazipur, Bangladesh

---

## [Decision Letter · Decision Letter 2]

16 Mar 2026

Green synthesis and characterization of zinc chitosan nanoparticles with their anti-bacterial study against rice pathogen Xanthomonas oryzae pv. oryzae

PONE-D-25-47822R2

Dear Dr. Latif,

We’re pleased to inform you that your manuscript has been judged scientifically suitable for publication and will be formally accepted for publication once it meets all outstanding technical requirements.

Kind regards,

Kandasamy Ulaganathan

Academic Editor

PLOS One

Additional Editor Comments (optional):

Reviewers' comments:

Reviewer's Responses to Questions

**Comments to the Author**

Reviewer #2: All comments have been addressed

2. Is the manuscript technically sound, and do the data support the conclusions?

Reviewer #2: Yes

3. Has the statistical analysis been performed appropriately and rigorously?

Reviewer #2: Yes

4. Have the authors made all data underlying the findings in their manuscript fully available?

Reviewer #2: Yes

5. Is the manuscript presented in an intelligible fashion and written in standard English?

Reviewer #2: Yes

Reviewer #2: (No Response)

.

Reviewer #2: No

---

## [Editor Report · Acceptance letter]

PONE-D-25-47822R2

PLOS One

Dear Dr. Latif,

I'm pleased to inform you that your manuscript has been deemed suitable for publication in PLOS One. Congratulations! Your manuscript is now being handed over to our production team.

Kind regards,

on behalf of

Dr. Kandasamy Ulaganathan

Academic Editor

PLOS One